# Connectomic analysis reveals an interneuron with an integral role in the retinal circuit for night vision

Silvia JH Park[1†], Evan E Lieberman[2†], Jiang-Bin Ke[2], Nao Rho[2], Padideh Ghorbani[2], Pouyan Rahmani[1], Na Young Jun[1], Hae-Lim Lee[3], In-Jung Kim[1], Kevin L Briggman[4], Jonathan B Demb[1,3,5]*, Joshua H Singer[2]*

[1]Department of Ophthalmology & Visual Science, Yale University, New Haven, United States; [2]Department of Biology, University of Maryland, College Park, United States; [3]Department of Cellular & Molecular Physiology, Yale University, New Haven, United States; [4]Circuit Dynamics and Connectivity Unit, National Institute of Neurological Disorders and Stroke, National Institutes of Health, Bethesda, United States; [5]Department of Neuroscience, Yale University, New Haven, United States

*For correspondence:
jonathan.demb@yale.edu (JBD);
jhsinger@umd.edu (JHS)

†These authors contributed
equally to this work

Competing interests: The
authors declare that no
competing interests exist.

Reviewing editor: Fred Rieke,
University of Washington, United
States

**Abstract** Night vision in mammals depends fundamentally on rod photoreceptors and the well-studied rod bipolar (RB) cell pathway. The central neuron in this pathway, the AII amacrine cell (AC), exhibits a spatially tuned receptive field, composed of an excitatory center and an inhibitory surround, that propagates to ganglion cells, the retina's projection neurons. The circuitry underlying the surround of the AII, however, remains unresolved. Here, we combined structural, functional and optogenetic analyses of the mouse retina to discover that surround inhibition of the AII depends primarily on a single interneuron type, the NOS-1 AC: a multistratified, axon-bearing GABAergic cell, with dendrites in both ON and OFF synaptic layers, but with a pure ON (depolarizing) response to light. Our study demonstrates generally that novel neural circuits can be identified from targeted connectomic analyses and specifically that the NOS-1 AC mediates long-range inhibition during night vision and is a major element of the RB pathway.

## Introduction

In dim light, vision originates with rod photoreceptors, which are sensitive enough to respond to single photons (*Ingram et al., 2016*; *Pugh, 2018*). In mammals, rod output is conveyed to ganglion cells (GCs), the retinal projection neurons, via a specialized circuit: the rod bipolar (RB) cell pathway (*Demb and Singer, 2015*; *Famiglietti and Kolb, 1975*; *Field et al., 2005*; *Figure 1*). The central neuron in the RB pathway is the AII amacrine cell (AC), which receives excitatory, depolarizing input from the RBs at light stimulus onset, and is therefore an ON cell. The AII provides feedforward signals that simultaneously drive depolarizing and hyperpolarizing responses in ON and OFF GC types, respectively (*Figure 1*; *Murphy and Rieke, 2006*; *Strettoi et al., 1994*; *Strettoi et al., 1992*). The circuitry of the RB pathway efficiently transforms weak sensory inputs into neural responses with high signal-to-noise ratio (*Field et al., 2005*) capable of guiding behavior at visual threshold (*Smeds et al., 2019*).

Although well-studied, several uncertainties about RB pathway function remain (*Demb and Singer, 2012*). Most significantly, we do not understand the cellular basis for the spatial receptive field of the AII, which is composed of an excitatory center, driven by RB input, and an antagonistic, inhibitory surround. The surround is apparently generated by the activity of a spiking, GABAergic AC that makes direct inhibitory synapses with the AII, but the identity of this critical interneuron and

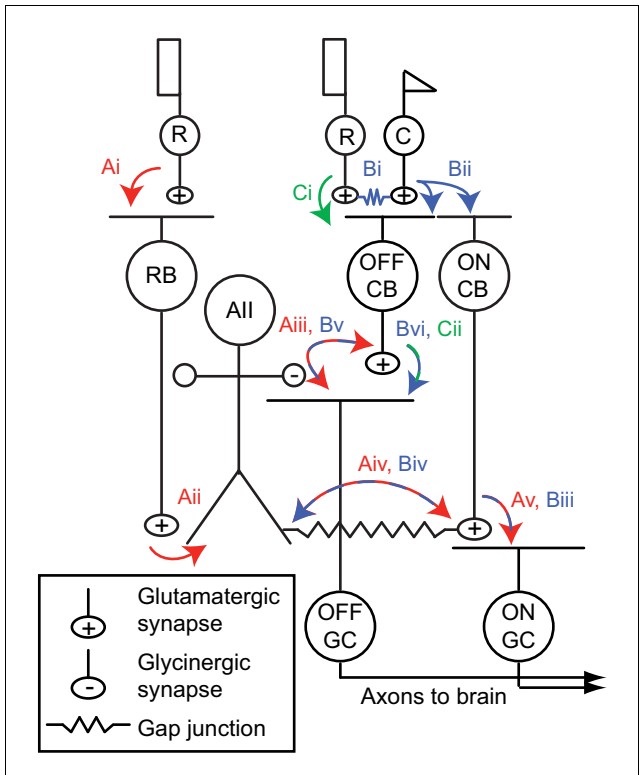

**Figure 1.** The mammalian retinal pathway for night vision. (Ai-Av) In red: the rod bipolar (RB) pathway of mammalian retina. Rods make synapses onto RBs (Ai), which make synapses onto the AII. Signaling from AIIs to ON and OFF pathways is compartmentalized by the morphology of the AII, a bistratified cell with distinct neurites in the ON and OFF sublaminae of the inner plexiform layer (IPL). (Aii). Proximal (OFF-layer) dendrites provide glycinergic synapses (Aiii) onto the terminals of some OFF cone bipolar (CB) cells [primarily type 2 CBs; (*Graydon et al., 2018*)] and onto the dendrites of some OFF ganglion cells (GCs), including OFF α and δ GCs as well as suppressed-by-contrast GCs (*Beaudoin et al., 2019*; *Demb and Singer, 2012*; *Jacoby et al., 2015*). At their distal (ON-layer) dendrites, AIIs are coupled by electrical synapses to the terminals of ON CBs (Aiv) (particularly type 6 CBs presynaptic to ON α GCs); depolarization of AIIs drives excitatory transmission to GCs, including ON α GCs (*Schwartz et al., 2012*) (Av). Thus, the AII mediates so-called 'cross-over' inhibition, whereby one pathway (ON, in this case) suppresses the other (OFF) and thereby decorrelates their outputs (*Demb and Singer, 2015*). (Bi-Bvi) In blue: rods are coupled electrically to cones by gap junctions (Bi), and cones make synapses onto ON and OFF CBs (Bii). Depolarization of the ON CB by the cone not only drives glutamatergic transmission to ON GCs (Biii), it also depolarizes AIIs via the electrical synapses (Biv) and thereby elicits glycinergic transmission to OFF GCs and perhaps OFF CBs (Bv). (Ci-Cii) In green: rods make direct chemical synapses onto some types of OFF CB (Ci), which in turn contact OFF GCs (Cii).

the retinal circuitry in which it participates remain unresolved (*Bloomfield and Xin, 2000*). Understanding mechanisms contributing to surround inhibition of the AII would explain how the retina encodes spatial information in dim light.

Here, we identified the major inhibitory input to the AII by combining anatomical, genetic, and electrophysiological analyses in a three-step process. One, we reconstructed ACs that provided synaptic input to AIIs in a volume of mouse retina imaged by scanning block-face electron microscopy [SBEM; (*Denk and Horstmann, 2004*)]. Two, we evaluated published descriptions of reporter mouse lines to identify genetically-accessible ACs that had the anatomical characteristics of the cells reconstructed from SBEM images. And three, we used electrophysiological recordings and genetics-based circuit analysis to demonstrate that a candidate AC, which provided the great majority of the inhibitory synaptic input to AIIs, exhibited a light response predicted by its anatomy and made GABAergic synapses onto AIIs. This spiking, GABAergic AC is a multistratified, ON AC denoted NOS-1 AC and identified in the nNOS-CreER mouse line (*Zhu et al., 2014*). We conclude that the NOS-1 AC is an integral component of the RB pathway and a significant source of long-range inhibition during night

vision. More generally, our study demonstrates the utility of targeted, small-scale 'connectomic' analysis for identification of novel neural circuits.

## Results

### AIIs in the mouse retina exhibit a TTX-sensitive, GABAergic receptive field surround

The inhibitory receptive field surround of AIIs has been studied extensively in the rabbit retina, where it is GABAergic and appears to be generated by spiking ACs because it is suppressed by the voltage-gated sodium channel blocker tetrodotoxin (TTX) (**Bloomfield and Xin, 2000**; **Xin and Bloomfield, 1999**). Here, we began by probing for the existence of a similar TTX-sensitive surround mechanism in AIIs of the mouse retina (**Figure 2A**).

AIIs were targeted for recording in the whole-mount retina (see Materials and methods; **Mortensen et al., 2016**), and rods were stimulated with light spots of varying diameter eliciting 10 photoisomerizations (R*)/rod/s, an intensity that reliably activates the RB pathway (see Materials and methods). Evoked excitatory currents (voltage-clamp; $V_{hold} = E_{Cl} = -70$ mV) increased with spot diameter well beyond the physical ~30 μm width of the AII dendritic field (**Figure 2B**). This wide receptive field of excitation is explained by electrical coupling within the AII network (**Hartveit and Veruki, 2012**; **Xin and Bloomfield, 1999**): an excitatory current originating in surrounding AIIs spreads laterally as a coupling current. Recording at $V_{hold} = E_{cation} = +5$ mV during spot presentation in control (Ames') medium yielded an evoked current comprising a mixture of genuine inhibitory current and unclamped coupling current. In the presence of TTX, the inhibitory input was suppressed, leaving the coupling current (**Figure 2A**). The difference current (Ames' – TTX; **Figure 2A**) is the isolated, TTX-sensitive, rod-driven inhibitory input, which increased as a function of spot diameter (**Figure 2C**). We observed as well that TTX exerted a mild suppressive effect on the excitatory light-evoked current, suggesting that spiking interneurons influence synaptic transmission from the presynaptic bipolar cells (**Figure 2B**).

### Propagation of the AII surround to downstream GCs

The surround suppression of AIIs is expected to propagate to the GCs that receive input from the RB-AII network. This includes three GC types that receive input from the RB pathway at or near visual threshold: the ON α GC and the OFF α and δ GCs, which exhibit high sensitivity to spatial contrast in the visual scene—similar to primate parasol cells—and project to the geniculo-cortical pathway (**Ala-Laurila et al., 2011**; **Ala-Laurila and Rieke, 2014**; **Dunn et al., 2006**; **Grimes et al., 2018a**; **Grimes et al., 2015**; **Grimes et al., 2014b**; **Grimes et al., 2018b**; **Ke et al., 2014**; **Kuo et al., 2016**; **Murphy and Rieke, 2006**; **Murphy and Rieke, 2008**). We recorded from these GC types while rods were stimulated with dim (evoking either 4 or 40 R*/rod/s) spots of varying size; this stimulus will evoke excitatory postsynaptic currents (EPSCs) in ON α GCs and inhibitory postsynaptic currents (IPSCs) in OFF α and δ GCs that reflect the output of the RB-AII network (**Murphy and Rieke, 2006**; **Murphy and Rieke, 2008**).

As spot diameter increased up to 2000 μm, EPSCs in ON α GCs and IPSCs in OFF α and δ GCs first increased and then decreased in amplitude, reflecting an initial increase in the receptive field center response and then subsequent surround suppression of the center response (**Figure 2D–F**). The IPSCs recorded in OFF α and δ GCs were similar, although not identical, likely reflecting shared inputs from the AII combined with cell type-specific inhibitory inputs from unique AC types. Nevertheless, in all three GC types, surround suppression was blocked similarly by TTX, suggesting that it was mediated by a common presynaptic mechanism: inhibition of the AII. Thus, the mouse AII exhibits a TTX-sensitive receptive field surround mediated by direct inhibitory synapses and this surround is propagated to GCs. To understand the mechanism for surround inhibition, we searched for the inhibitory ACs presynaptic to the mouse AII.

### Anatomical identification of inhibitory synaptic inputs to AIIs

We began by skeletonizing three AII ACs distributed across serial block face electron microscopy (SBEM) volume k0725 [mouse retina; 50 × 210 × 260 μm; (**Ding et al., 2016**); available at https://webknossos.org/datasets/Demo_Organization/110629_k0725/view]. AIIs were traced from their

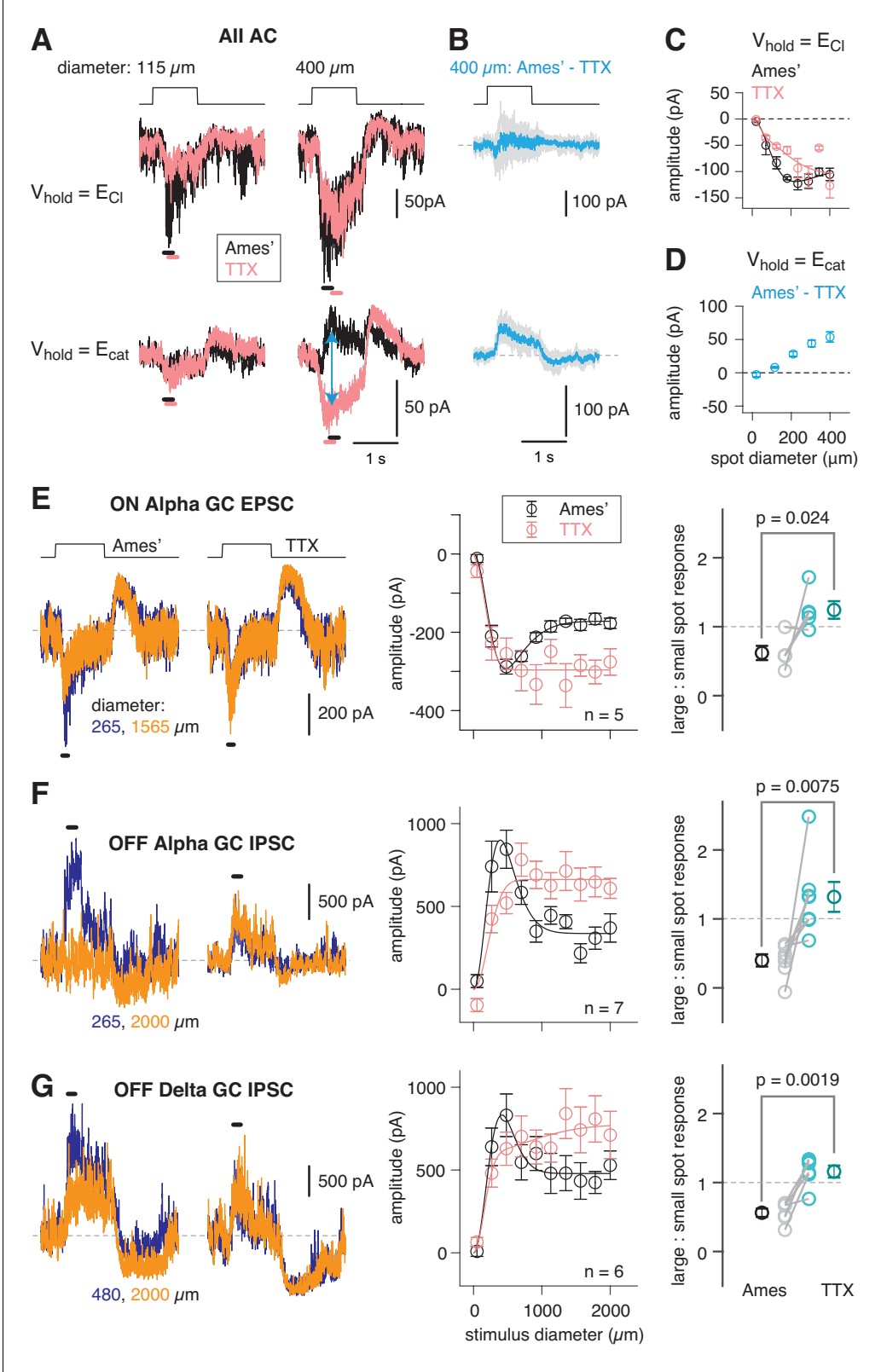

**Figure 2.** An inhibitory surround recorded in mouse AII ACs. (**A**) AII responses to spots of light (10 R*/rod/s, 1 s duration) with the indicated diameter in control condition (Ames' medium) and after applying TTX (1 μM). Responses were recorded at holding potentials ($V_{hold}$) near $E_{Cl}$ (−70 mV; top row) and near $E_{cat}$ (+5 mV; bottom row). (**B**) Difference current at each $V_{hold}$ for the 400 μm spot (average of n = 5 cells; shaded areas

*Figure 2 continued on next page*

Figure 2 continued

are ± SEM across cells as a function of time). (**C**) Response amplitude (measured over a 200 ms window: horizontal bars in **A**) to spot of variable diameter at $V_{hold}$ = $E_{Cl}$ (n = 5 cells). Error bars are ± SEM across cells. (**D**) Amplitude of difference current at $V_{hold}$ = $E_{cat}$ averaged across cells. Conventions as in (**C**). (**E**) Left, light flash (dark background, two spot sizes, 4 or 40 R*/rod/ s across cells)-evoked EPSCs in an ON α RGC. Under control conditions, the response is smaller for the larger diameter, illustrating the surround effect, which is blocked by TTX (1 μM). Middle, Spot stimuli of varying diameter (Ames' and TTX) elicit EPSCs; peak response amplitude measured in a time window (100–200 ms; horizontal line at left). Right, ratio of response to large (averaged over the three largest diameters) and small (chosen as the optimal spot size for each cell in the control condition) spots. A ratio <1 indicates a surround effect. The ratio increased significantly in TTX. (**F**) Same as (**E**) for IPSCs in OFF α RGCs (spot intensity, 4 R*/rod/s). (**G**) Same as (**F**) for OFF δ RGCs.

locations presynaptic to dyad ribbon synapses at RB terminals that were themselves identified by morphology and position within the inner plexiform layer (IPL) (*Graydon et al., 2018*; *Mehta et al., 2014*; *Pallotto et al., 2015*). For each of these three AIIs, we annotated all the inputs from ribbon synapses, arising from RBs, as well as all the conventional synaptic inputs, presumed to arise from inhibitory ACs (*Figure 3A*). AIIs received substantial input to their distal (ON layer) dendrites from RBs (*Table 1*; *Figure 3B*), consistent with published descriptions (*Strettoi et al., 1992*; *Tsukamoto and Omi, 2013*). With regard to inhibitory input, it is notable that virtually all AC inputs to the AII were in the inner IPL (ON layer), on the distal dendrites; the remainder of the AC inputs were on the somas and most proximal portions of the AII dendrites (*Table 1*; *Figure 3B*).

For each AII, we skeletonized 21 of the AC inputs to the distal dendrites to assess the morphology of the presynaptic neurons (*Figure 3C1*, left, and 3C2). Of the 63 AC skeletons created, 61 were of neurites, generally unbranched, that extended through the volume and appeared to be axons: each of these originated from an AC not contained in the SBEM volume (*Figure 3C2*). After annotating their output synapses, we determined that these axons made synapses with AIIs almost exclusively; the remainder of the output was to RBs with very few synapses to ON CBs and unidentified cells (*Table 2*; *Figure 3C1*, left, and 3C2). This determination was made by tracing the postsynaptic neurites sufficiently to identify RBs from their characteristic axon terminals, which are large and make dyad synapses with presumed AIIs and A17 ACs, and to identify AIIs based on several characteristic features: a soma position at the border of the INL and IPL; very thick proximal dendrites; and a postsynaptic position at RB dyad synapses (see *Graydon et al., 2018*; *Mehta et al., 2014*; *Strettoi et al., 1990*; *Strettoi et al., 1992*).

These 61 axons made 130 synapses onto the three reconstructed AIIs, giving rise to ~25% of the total inhibitory input to each AII. Therefore, assuming that each axon arises from a distinct cell, most AIIs receive ~2 inputs (i.e. ~130 synapses/61 axons) from each presynaptic AC; these inputs occur mostly (~80%) within 4 μm of RB→AII synapses (*Figure 3B and C*), with a median distance of 2.3 μm (*Figure 3C4*). Thus, the inhibitory AC inputs to the AII are well positioned for shunting local excitatory RB input.

The inputs to the AII somas and proximal (OFF layer) dendrites (*Figure 3B and C1*, right) were not considered in as much detail because these were few in number and presumed to arise from dopaminergic ACs (DACs) (*Contini and Raviola, 2003*; *Gustincich et al., 1997*; *Voigt and Wässle, 1987*). Indeed, when we traced 8 neurites presynaptic to the soma and proximal dendrites of a single AII, we identified 113 output synapses in total, 90 (80%) of which were to neighbor AIIs and 23 to other cells (*Figure 3C5*). The AII soma appeared to be enveloped in a 'basket' of such neurites (*Figure 3C6*), as noted previously for DAC→AII synapses (*Gustincich et al., 1997*; *Voigt and Wässle, 1987*).

Two of the 63 AC skeletons that made inputs to the AII distal dendrites could be traced to a relatively complete neuron contained within the SBEM volume (*Figure 4A,B*). These two ACs appeared to be of the same type and were characterized as displaced (with somas in the GC layer), multi-stratified cells, with processes in both the ON and OFF strata of the IPL.

## Anatomical assessment of an AC circuit presynaptic to the AII

We annotated all of the synaptic connections—inputs and outputs—made by the two ACs contained within the volume and then identified each pre- or postsynaptic partner (*Figure 4A,B*). Both ACs

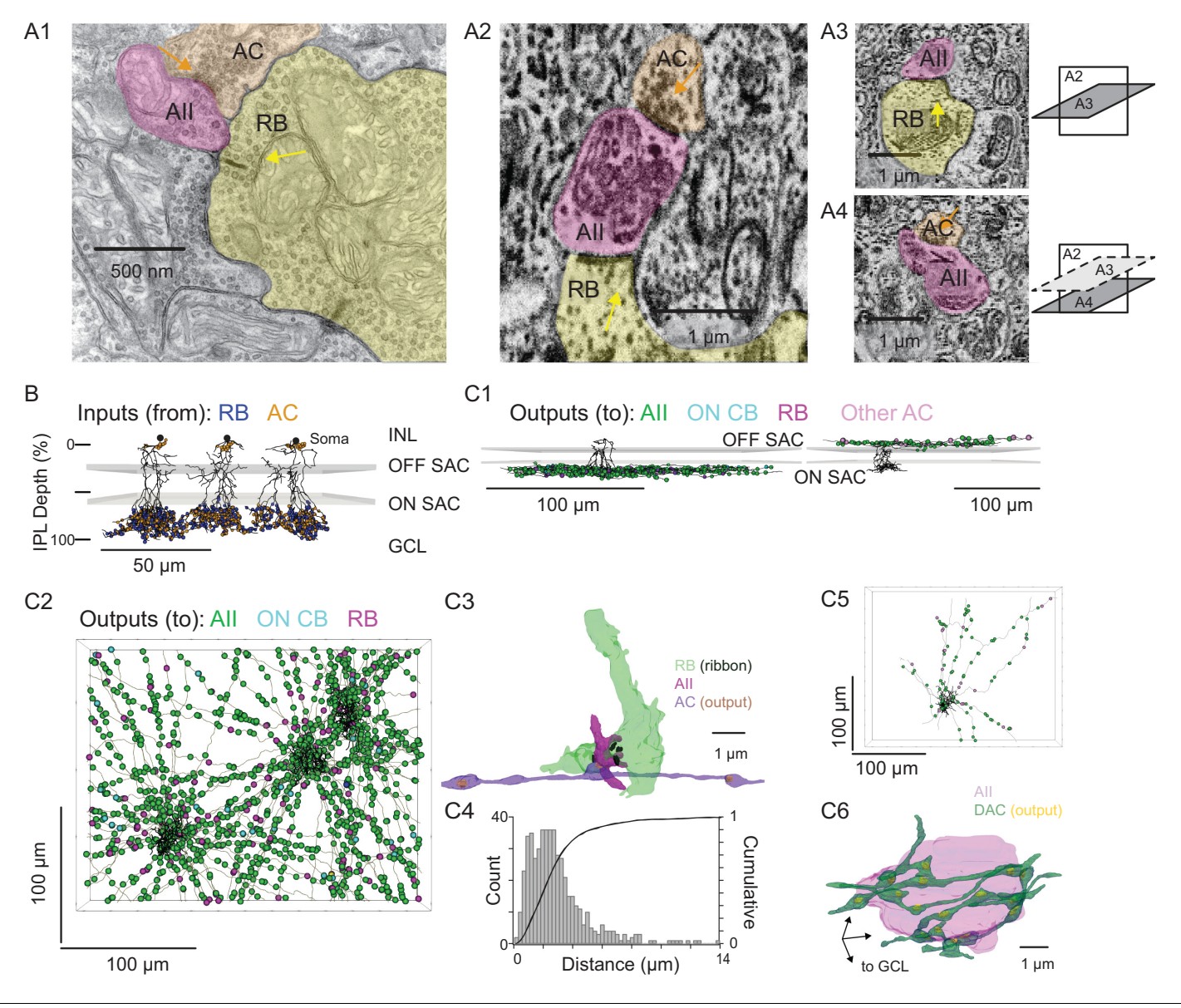

**Figure 3.** Anatomical characterization of AC axons presynaptic to the AII. (**A1**) A RB dyad synapse, visualized by TEM. Note the synaptic ribbon (yellow arrow) and the nearby AC input to the postsynaptic AII (orange arrow). (**A2**) A similar dyad synapse observed by SBEM. Again, the RB ribbon and AC input to the postsynaptic AII are marked with yellow and orange arrows, respectively. (**A3, A4**) Orthogonal views of the synapse illustrated in A2 showing the RB dyad (top) and the AC input to the postsynaptic AII (bottom). A3 and A4 are from different sections to illustrate the appearance of both ribbon and AC inputs in the section orthogonal to A2. This is illustrated schematically by the squares at right. (**B**) Three AIIs, skeletonized and annotated with either RB ribbon synapse locations or AC synapse locations. The view is from the side of the volume, representing a transverse section through the retina. The planes containing ON and OFF SAC dendrites are represented by gray rectangles. Note that the majority of AC inputs to the AII are found in the same IPL sublaminae as the RB inputs, the vitreal side of the ON SAC dendrites (from *Ding et al., 2016*). As well, AIIs receive no synaptic input from ACs in the IPL sublaminae between the ON and OFF SAC dendrites. AII skeletons and annotations are contained within *Source data 1* and downloadable in Knossos XML format. SAC skeletons are downloadable as *Source data 3*. (**C1**) Side view of a single AII and presynaptic neurites with output synapses annotated (right: presynaptic to distal dendrites; left: presynaptic to soma and proximal dendrites; different AIIs are shown in the two panels). (**C2**) An en face view, visualized from the GCL, of 3 AIIs (black) and 61 presynaptic axons. Note the preponderance of green circles, indicating synapses to the three reconstructed AIIs and other AIIs within the volume. Virtually every synapse that was not made with an AII was onto a RB. We observed two synapses onto these axons. Neurite annotations are contained within *Source data 1* and downloadable in Knossos XML format. (**C3**) Segmentation of the RB-AII-AC complex. The AC axon is thin, with occasional large varicosities containing clusters of vesicles. It makes synapses with AIIs, usually quite close to RB→AII ribbon synapses. In this example, the AII dendritic segment receives four ribbon inputs from the axon terminal varicosity of a RB and one conventional synaptic input from an AC. The other synapses made by this AC axon segment also are with

*Figure 3 continued on next page*

Figure 3 continued

AIIs. (C4) Conventional and normalized cumulative histograms of distribution of nearest-neighbor distances between AC→AII and RB→AII synapses. (C5) An *en face* (from GCL) view of a single AII and neurites presynaptic to its soma and proximal dendrites. (C6) Segmentation of an AII soma and presynaptic neurites, with presynaptic active zones annotated. The image is a tilted side view; the orientation axis (lower left) indicates the relative position of the GCL.

exhibited very similar patterns of connectivity, as quantified in *Table 2*. Interestingly, the vast majority of synaptic output was to AIIs (AII identity confirmed as described above); a smaller but significant portion was to RBs (again, confirmed as above), with the remainder to ON CBs (which were skeletonized completely; below). We did not observe any synapses onto other ACs or onto GCs. Thus, these two ACs appear to be representatives of a singlecell type that contacts preferentially neurons in the RB pathway, specifically AIIs. Given the similarity of the postsynaptic target neuron populations of these ACs and the axons reconstructed partially and described above (*Figure 3B*, left, and 3C2, 3), we believe all to be representative of the same AC type, a cell that contacts AIIs preferentially and provides the vast majority of inhibitory synapses to the AII.

Both ACs received very similar numbers of inputs from conventional and ribbon synapses (*Table 2*). The conventional synapses were presumed to arise from ACs; these synapses were annotated, but we did not attempt to identify the presynaptic cells apart from following their processes for a short distance through the volume to ensure that they indeed were ACs (e.g. they lacked presynaptic ribbons and made other conventional synapses). We presume that these conventional synapses are inhibitory, given that most ACs are either GABAergic or glycinergic, but given the existence of at least one type of excitatory AC—the VGLUT3 AC (*Johnson et al., 2004*; *Kim et al., 2015*; *Lee et al., 2016*)—it is possible that some subset are excitatory. The VGLUT3 AC itself, however, is unlikely to be a presynaptic partner given that its synaptic inputs and outputs are confined to the central portion of the IPL, whereas the dendrites of the two reconstructed ACs were at the inner and outer margins of the IPL (*Hsiang et al., 2017*; *Johnson et al., 2004*). Synaptic inputs both from *en passant*, axonal ribbon synapses and from more typical, axon terminal ribbon synapses were observed; analysis of the presynaptic cells revealed them to be ON bipolar cells—both RBs and ON CBs—exclusively, and the synapses therefore are glutamatergic (*Table 2*). Thus, this AC appears to receive only excitatory ON bipolar input despite its being a multi-stratified cell with processes in both ON and OFF sublaminae of the IPL.

The 121 ON CB synapses onto the 2 ACs arose from 67 presynaptic ON CBs, and we skeletonized these ON CBs in order to identify them based on morphological characteristics and axon terminal depth within the IPL (*Figure 4C*; *Ding et al., 2016*; *Helmstaedter et al., 2013*; *Manookin et al., 2008*; *Sabbah et al., 2017*; *Stabio et al., 2018*). Most of the presynaptic ON CBs belonged to the type 6 population; the others were a mix of types 5, 7 and 8 CBs (two cells could not be fully reconstructed and were unidentified) (*Figure 4C*). Fifteen *en passant* (axonal) synapses onto the outer (OFF-layer) dendrites of the reconstructed ACs were observed (*Figure 4D*): 13 from type 6 cells and 2 from the unidentified cells (likely type 6 CBs). Both the occurrence of these axonal ON bipolar cell synapses and their general appearance are consistent with previous reports (*Dumitrescu et al., 2009*; *Hoshi et al., 2009*; *Kim et al., 2012*; *Lauritzen et al., 2013*). Additionally, *en passant* synapses in axons of type 6 cells had a markedly distinct appearance from those in the other CB types: for *en passant* synapses in type 6 cells, ribbons tended to occur in clusters of at least three, all apposed to the same postsynaptic process (*Figure 4D*; *Figure 4—video 1*). In this respect, they resemble

**Table 1.** Inputs to AII amacrine cells.

|  | AII #1 | AII #2 | AII #3 | Mean ±SD |
|---|---|---|---|---|
| Inputs (from) |  |  |  |  |
| RB | 173 | 171 | 176 | 173 ± 3 |
| AC (Total) | 178 | 177 | 176 | 177 ± 1 |
| AC (ON layer) | 161 (96%) | 157 (89%) | 161 (91%) | 160 ± 2 (92 ± 4%) |
| AC (Soma) | 17 (4%) | 20 (11%) | 15 (9%) | 17 ± 3 (8 ± 4%) |

**Table 2.** Connetivity of ACs presynaptic to AIIs.

|  | Axons (*Figure 3C2*) | Cell 1 (*Figure 4*) | Cell 2 (*Figure 4*) |
|---|---|---|---|
| Inputs (from) | Total: 2 | Total: 314 | Total: 297 |
| AC |  | 200 (63.5%) | 183 (61%) |
| RB | 1 | 47 (15%) | 56 (19%) |
| ON CB | 1 | 66 (21%) | 55 (18.5%) |
| Unidentified |  | 2 (<1%) | 3 (1%) |
| Outputs (to) | Total: 1425 | Total: 115 | Total: 106 |
| AII | 1212 (85%) | 93 (81%) | 86 (81%) |
| RB | 173 (12%) | 11 (9.5%) | 18 (17%) |
| ON CB | 35 (2%) | 11 (9.5%) | 2 (2%) |
| Unidentified | 5 (<1%) |  |  |

strongly the axonal ribbons observed in calbindin-positive ON CBs of the rabbit retina, which are likely homologous to mouse type 6 CBs because both share similar morphology and make synapses with ON α GCs (*Hoshi et al., 2009*; *Kim et al., 2012*; *Schwartz et al., 2012*; *Tien et al., 2017*). Notably, *en passant* synapses were identified in the axons of some, but not all, reconstructed type 5, 7 and 8 ON CBs, indicating that such synapses are a common feature of ON CBs. None of these synapses, however, were presynaptic to the reconstructed ACs, suggesting that *en passant* synapses are formed with a high degree of target specificity.

The 103 RB synapses with these two ACs arose from 77 RBs (all of which were skeletonized; not shown). All RB→AC synapses were dyads (*Figure 4E*; *Figure 4—video 2*); at 75% of these dyads, the AC replaced the A17 (i.e. the other postsynaptic cell was an AII), and at the remainder of the dyads, the AC either replaced the AII (18%) or else the identity of the second postsynaptic cell could not be confirmed (7%). In at least two of the cases in which the second cell at the dyad could not be identified, the dendrite had the appearance of those of the skeletonized ACs: a very thick dendrite containing clear cytoplasm. Thus, it appears that this AC receives a significant portion of its excitatory input (48%; see *Table 2*) from the RB population but that any individual RB provides only one or two synapses to a single AC; the latter finding is consistent with the vast majority of RB output being at dyad synapses with AII and A17 amacrine cells, in agreement with the standard model of RB synaptic output (*Demb and Singer, 2012*).

A number of multistratified AC types have been reported in anatomical studies of the mouse retina (*Badea and Nathans, 2004*; *Lin and Masland, 2006*; *Pérez De Sevilla Müller et al., 2007*). The vast majority of these, however, differ in their morphology from the ACs studied here: for example of the 16 wide-field, axon-bearing ACs categorized by *Lin and Masland (2006)* (their Figure 10), none have axons in the inner (ON) layer of the IPL and a narrow field of dendrites in the outer (OFF) layer. We were struck, though, by the resemblance of the AC identified here to one identified in a screen of Cre-driver lines: the NOS-1 AC of the nNOS-CreER mouse (*Zhu et al., 2014*). Therefore, we tested the hypothesis that the NOS-1 AC is the spiking, ON AC that provides inhibitory synaptic input to the AII.

## The NOS-1 AC is a spiking ON cell

NOS-expressing (NOS+) ACs were targeted for in vitro recording by crossing nNOS-CreER mice with Cre-dependent reporter mice (Ai32: ChR2/eYFP fusion protein expression; *Zhu et al., 2014*; *Park et al., 2015*) and then inducing recombination in ~1 month old offspring by tamoxifen injection (see Materials and methods). eYFP fluorescence (eYFP+) observed by two-photon laser scanning microscopy (2PLSM) was used to target NOS+ ACs for recording. We studied retinas in which Cre expression was induced robustly; these had $110 \pm 5$ cells $\text{mm}^{-2}$ (mean ± SEM) labeled in the GCL and $128 \pm 5$ cells $\text{mm}^{-2}$ labeled in the INL (n = 7 retinas from 7 mice). Based on an earlier description of this driver line, we assumed that labeled cells in the GCL included a mixture of NOS-1 and NOS-2 ACs, with NOS-1 cells having the bistratified morphology described here and NOS-2 cells exhibiting thick, spiny dendrites that project into the central level of the IPL, between the layers marked by

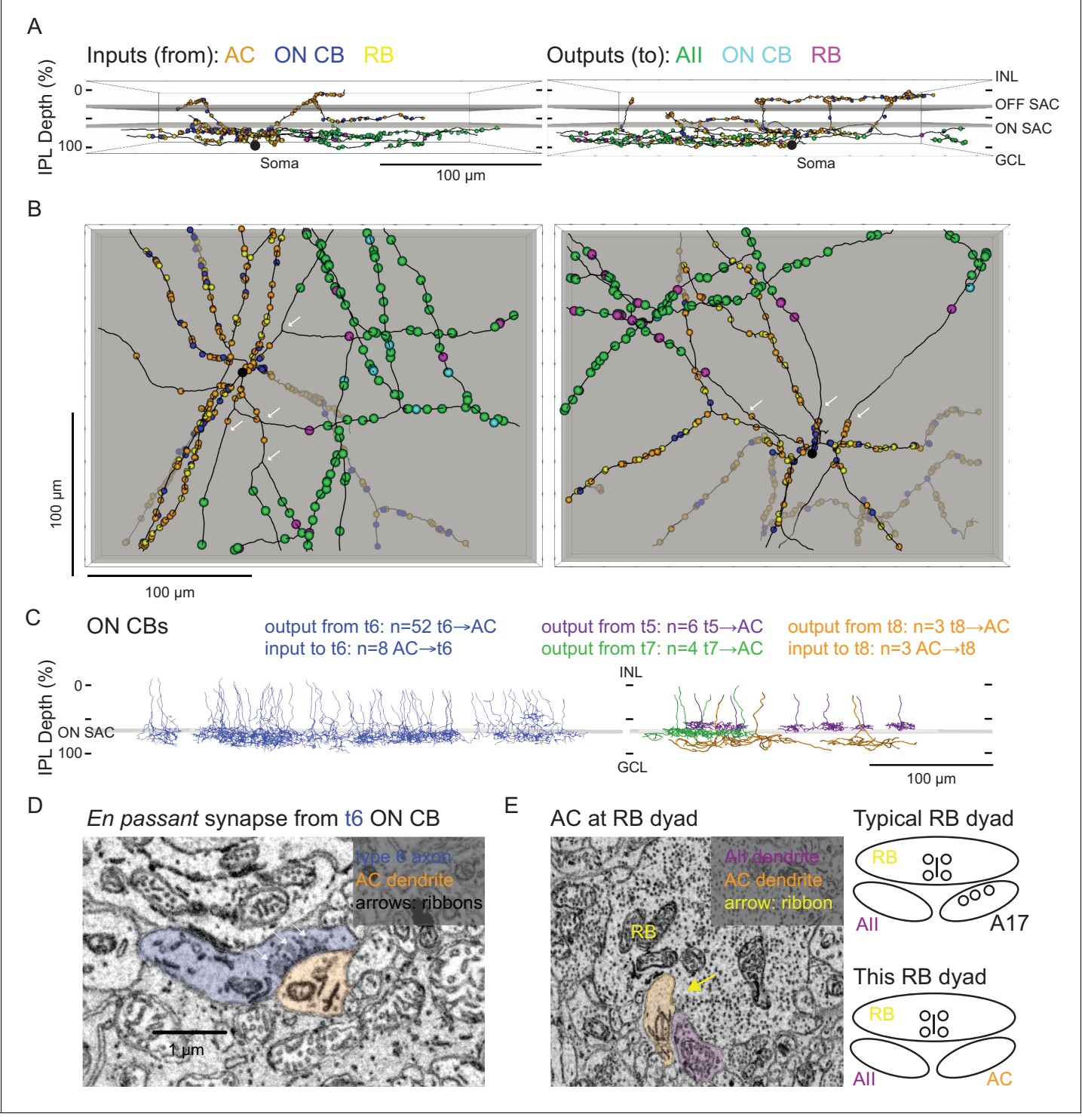

**Figure 4.** Anatomical characterization of an AC presynaptic to the AII. (**A**) Skeletonization and annotation of two ACs (left and right panels) with similar morphologies and patterns of synaptic connectivity. Note that both cells, viewed from the side, as in a transverse section through the retina, have similar neurite branching patterns and receive synaptic input from ACs and from ON CBs on dendrites in the OFF laminae of the IPL, on the outer side of the OFF SAC dendrites (i.e. close to 0% IPL depth; here, SAC dendrites are represented as gray rectangles). AC skeletons and annotations are contained within *Source data 1* and downloadable in Knossos XML format. (**B**) An *en face* view (viewed from the GCL; the gray represents the layer of ON SAC dendrites) of the two ACs illustrated in (**A**). Note that their synaptic inputs and outputs are segregated to different sections of their processes; the area receiving input is dendritic, and the area making output is axonal. White arrows indicate areas where dendrites become axons (inputs are proximal to the arrow, closest to the soma; outputs are distal to the arrow, farther from the soma). AC skeletons and annotations are contained within

*Figure 4 continued on next page*

*Figure 4 continued*

*Source data 1* and downloadable in Knossos XML format. (C) Side (transverse) view illustrating all ON CBs pre- or postsynaptic to the two ACs illustrated in (A) and (B). ON CBs were classified based on axon branching pattern and stratification depth relative to the ON SAC dendrites (*Helmstaedter et al., 2013*). CB skeletons and annotations are contained within *Source data 2* and downloadable in Knossos XML format. (D) Example *en passant* ribbon-type synapses in a type 6 ON CB axon. Note three ribbons clustered together and presynaptic to the same AC process. See *Figure 4—video 1* for a larger image stack. (E) Example of RB dyad at which the AC type shown in (A) and (B) replaces the A17 as one of the two postsynaptic cells (see schematic at right). See *Figure 4—video 2* for a larger image stack.

The online version of this article includes the following video(s) for figure 4:

**Figure 4—video 1.** Image stack containing *en passant* type 6 CB→AC synapses illustrated in *Figure 4D*.
https://elifesciences.org/articles/56077#fig4video1

**Figure 4—video 2.** Image stack containing RB→AII, AC dyad synapse illustrated in *Figure 4E*.
https://elifesciences.org/articles/56077#fig4video2

dendrites of ON and OFF starburst ACs (*Jacoby et al., 2018*; *Zhu et al., 2014*). Labeled cells in the INL included additional NOS-2 ACs (*Zhu et al., 2014*) as well as other AC types that projected into the outer most levels of the IPL but were not studied here.

Dye filling (Lucifer Yellow) of recorded eYFP+ cells in the GCL revealed that these cells most typically were NOS-1 ACs, with bistratified dendrites and long axons in the ON sublaminae identified by 2PLSM visualization following recording and, in some cases, by subsequent analysis by confocal microscopy (n = 13 cells; *Figure 5A*). All spiking cells that were imaged shared a similar morphology, indicating that they represent a single cell type (*Zhu et al., 2014*). We examined NOS-1 light responses using contrast modulation around a bright mean luminance ($10^4$ R* cone$^{-1}$ s$^{-1}$), because the IR laser exposure used to identify fluorescent cells in 2PLSM partially bleaches rods (see *Borghuis et al., 2013*; *Borghuis et al., 2011*). Under this condition, whole-cell recordings of membrane voltage (i.e. current-clamp recordings) showed that NOS-1 cells fired spikes in response to positive contrast and that spiking could be suppressed completely by negative contrast (*Figure 5B*). Responses increased in magnitude with increasing spot diameter, suggesting an integration area of at least ~500 µm diameter (*Figure 5C*).

NOS-2 cell membrane voltage responses to light clearly differed from those of NOS-1 cells (*Figure 5D,E*). NOS-2 cells were non-spiking with graded, depolarizing responses to both positive and negative contrast and are therefore ON-OFF cells (*Jacoby et al., 2018*). Both ON and OFF responses of NOS-2 cells increased with spot diameter, with a relatively more gradual increase for the OFF response (*Figure 5F*).

Voltage-clamp recording from NOS-1 cells demonstrated that positive contrast evoked excitatory synaptic input, measured as inward current relative to a baseline holding current ($V_{hold}$ = $E_{Cl}$) (*Figure 5G,H*). Negative contrast evoked a reduction of this baseline holding current, consistent with temporary suppression of ongoing presynaptic glutamate release (*Figure 5G,H,J*). Excitatory input was blocked completely by L-AP4, which suppresses the responses of ON bipolar cells, and therefore, excitation was mediated exclusively by the ON pathway (*Slaughter and Miller, 1981*; *Figure 5G*). Inhibitory synaptic input ($V_{hold}$ = $E_{cat}$) was measured at both positive and negative contrast, but the amplitude of the modulated current typically was small under control conditions. In the presence of L-AP4, however, inhibition persisted only in response to negative contrast and, indeed, increased by 38 ± 11.4 pA in amplitude (t = 3.3; n = 9; p<0.005; *Figure 5G–I*). This increase could reflect relief of ON pathway-mediated suppression of inhibitory input from OFF ACs to NOS-1 ACs. In summary, though, these experiments demonstrate conclusively that NOS-1 cells receive tonic glutamatergic input from ON bipolar cells that can be modulated by contrast as well as inhibitory inputs from ACs in both ON and OFF pathways.

## The NOS-1 AC is the predominant NOS-expressing AC in the ganglion cell layer and is distinct from CRH-expressing ACs

To determine the relative density of displaced NOS-1 and NOS-2 ACs in the GCL of the nNOS-CreER retina, we made loose patch recordings of responses to light from displaced eYFP+ ACs targeted by 2PLSM (*Figure 6A*). Most displaced ACs were spiking cells with sustained ON responses, confirming their identity as NOS-1 cells (n = 22/26 cells, three retinas from two mice). One exception was a spiking cell with both ON and OFF responses (*Figure 6A*, asterisk); we did not study this cell

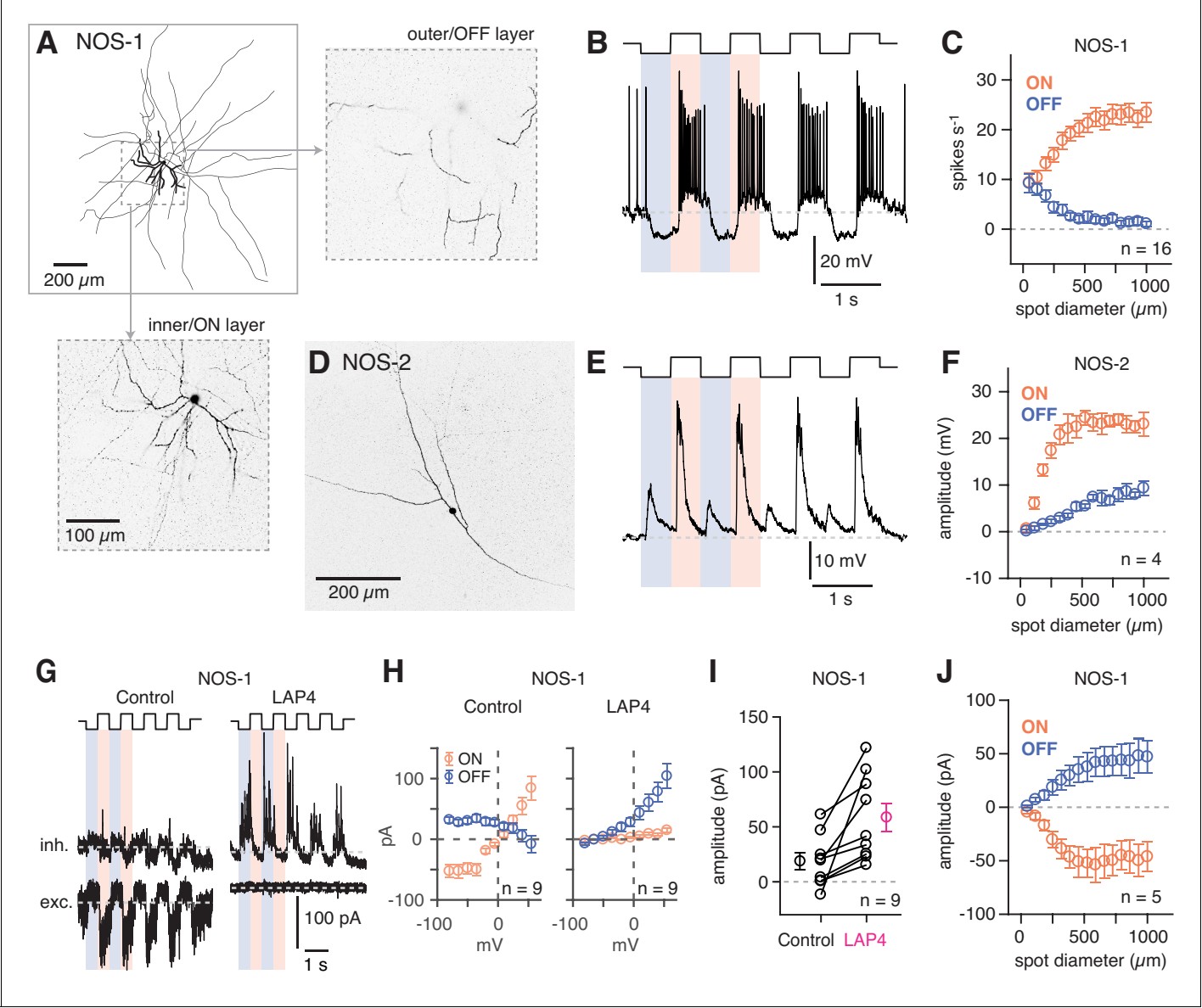

**Figure 5.** NOS-1 ACs are spiking ON-center cells that can be distinguished from NOS-2 ACs. (**A**) Morphology of a NOS-1 AC. Dendrites (thick) and axons (thin) were drawn from confocal images of a NOS-1 AC filled by Lucifer Yellow during whole-cell recording. Single confocal sections are shown for the inner/ON and outer/OFF layers of processes for the region indicated (dashed, boxed region). The cell is bistratified in the region proximal to the cell body; only the ON-layer processes are shown in the drawing. (**B**) Membrane potential recording for the cell in (**A**). The cell had a baseline firing rate at mean luminance (~$10^4$ R*/cone/s) that modulated above and below baseline during positive and negative contrast periods, respectively (spot diameter, 600 µm; 100% contrast). (**C**) Population (n = 16 cells) changes in firing rate during positive contrast (ON response) and negative contrast (OFF response) as a function of spot size (100% contrast). Firing rate was computed over a 500 ms time window for each contrast. Error bars indicated ± SEM across cells. (**D**) Collapsed confocal stack (maximum projection image) of a filled NOS-2 cell. (**E**) The NOS-2 cell in (**D**) responds with depolarization at both positive and negative contrast, an ON-OFF response (spot diameter, 600 µm; 100% contrast). (**F**) Population (n = 4 cells) changes in membrane potential as a function of spot size (measured over a 100 ms time window). Conventions are the same as in (**C**). (**G**) Excitatory and inhibitory current measured in a NOS-1 cell to a spot stimulus (diameter, 400 µm). After adding L-AP4 (20 µM) to suppress the ON pathway, the excitatory current is blocked and the inhibitory current increases and is OFF responding. (**H**) Current-voltage (I–V) plots for ON and OFF responses for data in (**G**) averaged across cells (measured over a 100- to 200-ms time window). Error bars indicate ± SEM across cells. (**I**) Population change in the OFF inhibitory current after adding L-AP4. Individual cell data are connected by lines. Population data indicate mean ± SEM across cells. (**J**) Excitatory current amplitude for a population of NOS-1 cells (n = 5 cells) as a function of spot diameter (100% contrast; measured over a 100 ms time window). Error bars indicate ± SEM across cells.

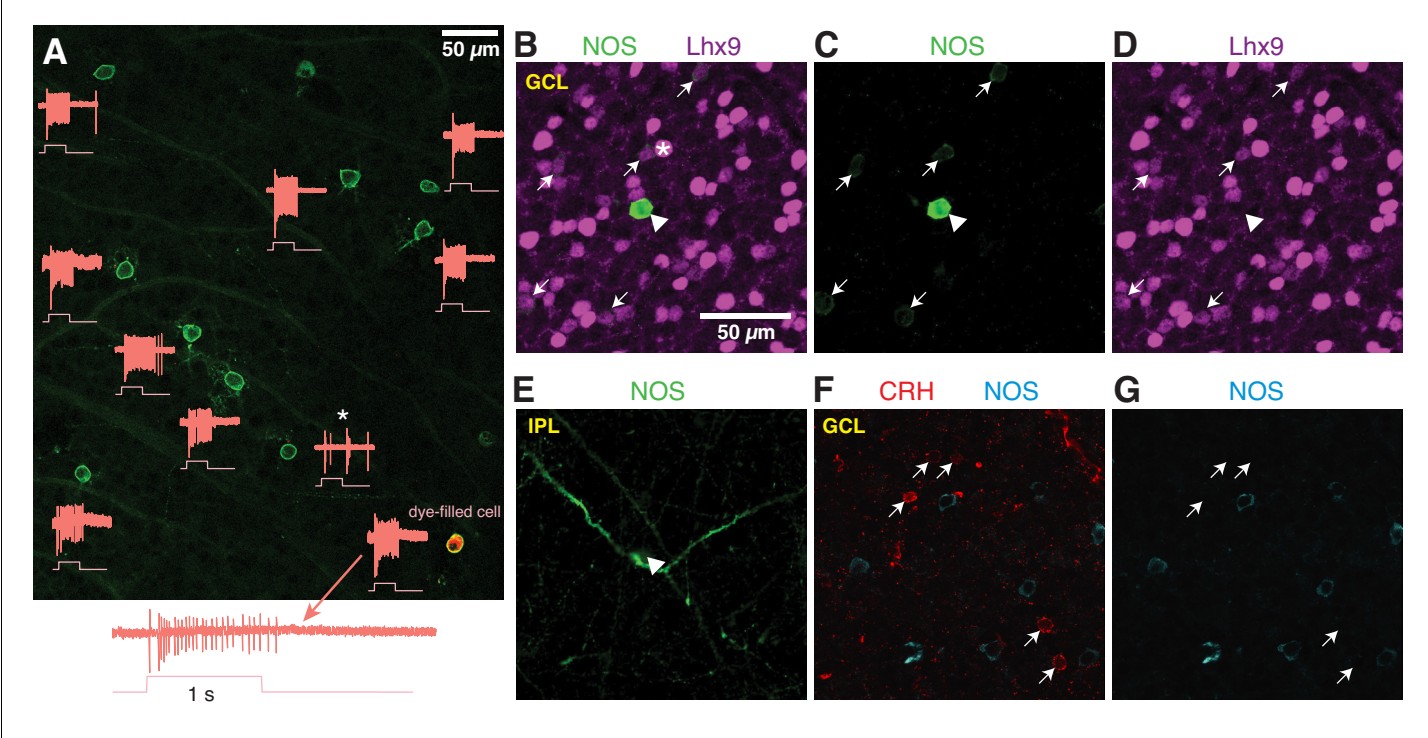

**Figure 6.** NOS-expressing ACs in the ganglion cell layer are primarily NOS-1 cells. (**A**) Loose-patch spike recording from a region of retina with cells labeled in the nNOS-CreER x Ai32 mouse. The response to a light flash (800-μm diameter, ~$10^4$ R*/cone/s) is shown next to each soma that was recorded. The majority of cells showed a sustained ON response in the spike rate during the light flash. In one case, the response differed (*) and showed a transient ON-OFF response. The cell at lower right was subsequently studied by whole-cell recording and was filled with dye; for this cell, the spike response is shown at an expanded scale below the image. (**B–E**) P12 retinas (C57/B6 wild-type) stained with antibodies to label NOS and Lhx9 expressing cells in the GCL. Most NOS-expressing cells showed dim labeling for Lhx9 antibody (arrows). A well-stained NOS-expressing cell in the center (arrow-head) did not show Lhx9 labeling. Laser power was increased for Lhx9 imaging, such that fluorescence of strongly-labeled cells was saturated (*, example cell), making it easier to visualize weakly-labeled cells. NOS labeling alone (**C**) and Lhx9 labeling alone (**D**). (**E**) Same as (**C**) with the image plane shifted to the inner plexiform layer (IPL). The NOS-expressing cell that lacked Lhx9 expression (arrowhead) had thick dendrites that could be followed into the IPL, with the characteristic properties and stratification of a NOS-2 cell. (**F–G**) P12 retina (C57/B6) stained with antibodies to label corticotropin releasing hormone (CRH) and NOS. NOS-expressing cells do not overlap with CRH-expressing cells. NOS labeling alone (**G**). Scale bar in (**B**) applies to (**B**) – (**G**).

further, but it likely represents either a subtype of nNOS AC that is primarily in the INL and only rarely displaced to the GC or a case of ectopic Cre and/or reporter expression. In the NOS-1 population, ~2–3 cells could typically be found within ~54,500 μm$^2$, the cross-sectional area of our SBEM volume; this cell density is consistent with our finding two putative NOS-1 cells in the SBEM data set.

Additionally, we examined NOS-expressing cells by nNOS antibody labeling during a developmental period (P8 – P14) during which a subset of NOS+ cells express the transcription factor Lhx9 (**Balasubramanian et al., 2018**). Two populations of NOS+ cells were observed: a small group of brightly-labeled cells (n = 19 cells) and a much larger group of dimly-labeled cells (n = 550 cells; six retinas from six mice; n = 2 retinas each at P8, P12 and P14). In retinas double-labeled with the Lhx9 antibody (two retinas from two animals, P12), only the dimly-labeled NOS+ cells were Lhx9+ (**Figure 6B–D**) (n = 146/156 cells). The brightly-labeled, Lhx9- cells (n = 5/5 cells, two retinas from two mice) exhibited dendrites that could be followed into the center of the IPL, where they gave rise to the thick, spiny processes characteristic of NOS-2 cells (**Figure 6E**). We conclude that Lhx9 expression distinguishes presumptive NOS-1 from NOS-2 cells in the GCL at P12, and that at all developmental time points, NOS-2 cells are rarely found in the GCL (3.3% of cells), making NOS-1 cells a large majority in the GCL (96.7% of NOS+ cells).

The NOS-1 cell was described also as the CRH-2 AC (Corticotropin-releasing hormone AC-2) based on labeling in the CRH-ires-Cre retina (*Zhu et al., 2014*). We, however, found that Cre-expressing cells in the CRH-ires-Cre::Ai32 retina are rarely labeled by the nNOS antibody (*Park et al., 2018*), suggesting limited overlap in CRH+ and NOS+ AC populations. Indeed, we found no co-expression of CRH and NOS by immunohistochemistry (P12 retina) (n = 66 CRH+ cells and 85 NOS+ cells, one retina) in neurons in the GCL (*Figure 6E–G*). Thus, NOS-1 ACs do not appear to co-express CRH, and we conclude, then, that Cre expression in NOS-1 ACs of the CRH-ires-Cre line is rare and unsystematic.

## Confirmation of synapses between NOS-1 cells and AIIs

To demonstrate that NOS-1 ACs provide synaptic input to AIIs, we eliminated Cre-expressing ACs in the nNOS-CreER retina by intraocular injection of an AAV containing a Cre dependent-DTA construct followed 2–3 days later by tamoxifen administration. Four weeks later, we stimulated the retina in vitro with dim (40 R*/rod/ s) spots of varying size and recorded light-evoked currents in AIIs at $V_{hold} = E_{Cl}$ or $V_{hold} = E_{cat}$ (*Figure 7A*; as in *Figure 2A*). Following ablation of NOS+ ACs (including both NOS-1 and NOS-2 cells), the recorded excitatory currents were smaller than in the control condition and were unaffected by TTX (*Figure 7B,C*; compare to *Figure 2B*), consistent with a loss of inhibitory input from NOS-1 ACs to presynaptic RBs (*Figure 4* and *Table 2*). Most significantly, though, we observed a lack of inhibitory TTX-sensitive surround in AII responses following NOS+ AC ablation (*Figure 7B,D*): the TTX-sensitive IPSCs evoked by large-diameter spots (590 to 800 µm) were significantly smaller in DTA-expressing retinas than in control mice that received the same AAV and tamoxifen injections but were Cre-negative.

We confirmed the elimination of NOS+ ACs by nNOS antibody staining and found that the number of NOS+ cells was reduced strongly in DTA-expressing retinas relative to controls (*Figure 7E,G*). To confirm that DTA expression did not result in non-specific ablation of ACs, we used ChAT antibody immunolabeling to quantify the number of starburst ACs (in both the INL and GCL) in control and DTA-expressing retinas and found no reduction in the experimental group (*Figure 7F,H*). Thus, elimination of NOS+ ACs provides evidence that a NOS+ cell, most likely the NOS-1 AC, is necessary for generating the AII inhibitory surround.

To confirm that NOS-1 ACs provide synaptic input to AIIs, we performed optogenetic circuit-mapping experiments after inducing ChR2/eYFP expression in NOS+ cells, as described above. After blocking the influence of the photoreceptors (see Materials and methods), we recorded from cells in a retinal whole-mount preparation and stimulated ChR2-expressing neurons with bright blue light. A representative ChR2-expressing NOS-1 cell (n = 3 total) responded within a few ms of the optogenetic stimulus with increased spiking (*Figure 8A*); IPSCs recorded in AIIs (normalized to their peak amplitude, 39 ± 6 pA) were observed a few ms later (*Figure 8A*), consistent with a monosynaptic connection from NOS-1 cells.

ChR2-evoked IPSCs in AIIs were blocked by SR95531 (50 µM) and therefore were mediated by GABA$_A$ receptors (*Figure 8B,C*) (reduction of 34.5 ± 11.7 pA, or 108 ± 4%; t = 28.5; n = 4; p<0.001). In a second group of cells, the IPSCs were blocked by TTX (1 µM), indicating that they arose from a spiking presynaptic AC (*Figure 8D,E*) (reduction of 35.1 ± 8.9 pA or 105 ± 4%; t = 29.5; n = 5; p<0.001). These results are consistent with a direct synaptic input from the NOS-1 AC to the AII because input from the non-spiking NOS-2 AC would not be TTX-sensitive. The spiking dopaminergic AC (DAC) also makes GABAergic synapses with the somas and proximal dendrites of AIIs (*Figures 3B2*, C3–5) (*Gustincich et al., 1997*), but the DACs in the nNOS-CreER retina do not express ChR2/eYFP, as demonstrated immunohistochemically: there was no overlap between TH+ cells, identified by TH immunolabeling (n = 90 cells), and eYFP+ cells (n = 198 cells, five retinas from three mice) (*Figure 8F,G*).

Additionally, we recorded ChR2-evoked IPSCs in RBs in retinal slices (n = 2) to confirm the functionality of NOS-1 AC→RB synapses observed in our anatomical analyses (*Figures 3C1, C2, C6* and *4A,B*). Here, we had to include the K channel blockers 4-AP (1 µM) and TEA (10 µM) in the external solution to enhance the excitability of cut NOS-1 AC axons so that they could be stimulated adequately in the slice (*Figure 9*, left). Indeed, control experiments recording from AIIs in both slice (*Figure 9*, left) and whole-mount (*Figure 9*, right) retinal preparations demonstrated that this manipulation enhanced release from NOS-1 cells significantly (n = 3); IPSCs were blocked by pictotoxin (50

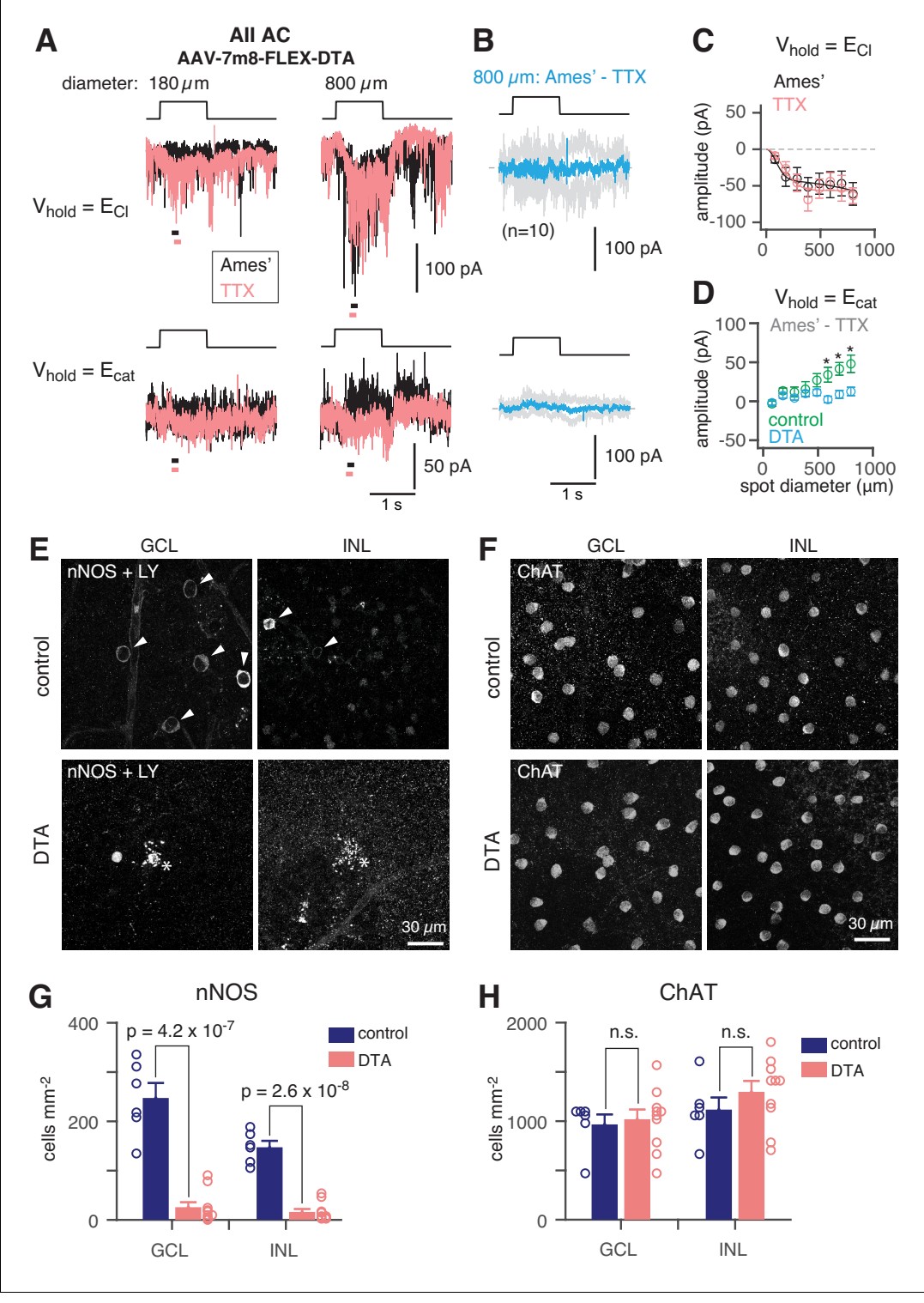

**Figure 7.** NOS+ ACs generate the TTX-sensitive receptive field surround of the AII. (**A**) AII responses to spots (diameter indicated; 40 R*/rod/s, 1 s duration) in control (Ames') and in TTX (1 μM) recorded at $E_{Cl}$ (−70 mV; top) and near $E_{cat}$ (+5 mV; bottom) in a nNOS-CreER retina injected with Cre-dependent DTA virus (AAV-7m8-FLEX-DTA). (**B**) Difference current at each $V_{hold}$ for the 800 μm diameter spot (average of n = 10 cells; shaded areas are ± SEM across cells as a function of time). (**C**) Spot (variable diameter) response amplitudes [measured over a 150 ms time window, indicated by horizontal bars in **A**] at $V_{hold} = E_{Cl}$ (n = 10 cells). Error bars are ± SEM across cells. (**D**) Difference current amplitude at $V_{hold} = E_{cat}$ averaged across cells. Same conventions as in (**C**). Recordings

*Figure 7 continued on next page*

*Figure 7 continued*

from control retinas are superimposed (n = 6 cells). Control mice had the same AAV and tamoxifen injections but were Cre-negative. Responses were smaller in the DTA group compared to the control group (one-tailed t-tests, *): 590 µm, t = 3.16, p=3.51×10$^{-3}$; 695 µm, t = 3.31, p=2.12×10$^{-3}$; 800 µm, t = 2.93, p=5.45×10$^{-3}$). (E) nNOS immunolabeling in GCL and INL, centered on a region with a recorded AII (visible in some images, marked by *). (F) Same format as (E) for ChAT immunolabeling of starburst ACs. (G) NOS+ cell density over a square region (0.64 × 0.64 mm) centered on a recorded AII and visualized by nNOS immunolabeling: DTA (n = 10 retinas) vs. control (n = 6 retinas). Virus-injected retinas had significantly fewer cells (one-tailed t-test): GCL, t = 8.35, p=4.2×10$^{-7}$; INL, t = 10.5, p=2.6×10$^{-8}$. (H) Same format as (G) for ChAT immunolabeling. Cell density assessed over a square region (0.16 × 0.16 mm) centered on a recorded AII. Starburst AC density in DTA virus-injected retinas (n = 10) was not significantly (n.s.) different than in control retinas (n = 6): GCL, t = −0.35; INL, t = −1.05.

µM). Notably, enhancing excitability in both retinal slices and whole-mount preparations did not change the latency of the IPSCs, supporting the conclusion that they are monosynaptic.

## Discussion

Here, we investigated an unresolved mechanism for the inhibitory, receptive field surround of the central interneuron in the RB pathway, the AII AC (*Figure 1*). Ultrastructural analysis of inhibitory input to three AIIs indicated that the majority of AC synapses arise from a single type of displaced, multistratified, wide-field cell that contacts AIIs preferentially. Two relatively complete examples of this AC were found within our SBEM volume, and examination of this cell type's outputs revealed that ~80% of its synapses were presynaptic to AIIs, with the majority of the remainder presynaptic to RBs. This AC received excitatory input exclusively from ON bipolar cells, both at *en-passant* axonal synapses in the OFF strata of the IPL and at axon terminal synapses in the ON strata of the IPL. The AC identified by SBEM analysis is a morphological match to the genetically identified NOS-1 AC, which we demonstrated to have physiological functions predicted by the ultrastructural analysis: it is an ON AC that provides GABAergic inhibition to AIIs and RBs. We therefore consider the NOS-1 AC to be an integral part of the well-studied mammalian RB pathway, serving as the major source of direct, long-range inhibition during night vision.

### Multiple cell types in nNOS-creER retina

Cre-mediated recombination drove ChR2/eYFP reporter expression in multiple AC types in the nNOS-creER retina. We observed two AC types, called NOS-1 and NOS-2, (*Jacoby et al., 2018*; *Zhu et al., 2014*), consistent with reports of nNOS expression in at least two AC types (*Chun et al., 1999*; *Kim et al., 1999*).

The NOS-1 AC is described above. The NOS-2 AC is a monostratified cell, with dendrites in the center of the IPL (between the processes of ON and OFF starburst ACs; *Figure 5D*); its soma is either in the conventional location in the INL (~75%) or displaced to the GCL (~25%) (*Chun et al., 1999*; *Jacoby et al., 2018*; *Kim et al., 1999*; *Zhu et al., 2014*). There were additional labeled ACs in the INL of the nNOS-CreER retina: their processes stratified in the OFF layers proximal to the INL, but the cells were not characterized further.

Our anatomical evidence, however, supports the conclusion that the NOS-1 AC is the only Cre-expressing AC presynaptic to the AII: NOS-2 dendrites are confined to the center of the IPL, between the OFF and ON starburst ACs, whereas inhibitory inputs to AIIs are either distal to the OFF starburst processes and apparently arise from dopaminergic ACs (not labeled in the nNOS-CreER line; *Figure 6F,G*) or proximal to the ON starburst AC processes (*Figure 3*). As well, in tracing many AC inputs to AIIs, composing ~25% of the total inhibitory synaptic input to these AIIs, we never encountered an axon that stratified between the ON and OFF starburst ACs.

### The NOS-1 AC is the dominant inhibitory input to the AII and generates the AII surround

The AII has an ON-center receptive field with an antagonistic surround (*Bloomfield and Xin, 2000*; *Nelson, 1982*; *Xin and Bloomfield, 1999*). The surround is mediated by GABAergic inhibition, it depends on activation of ON bipolar cells, and it is blocked by TTX, indicating that it arises from a

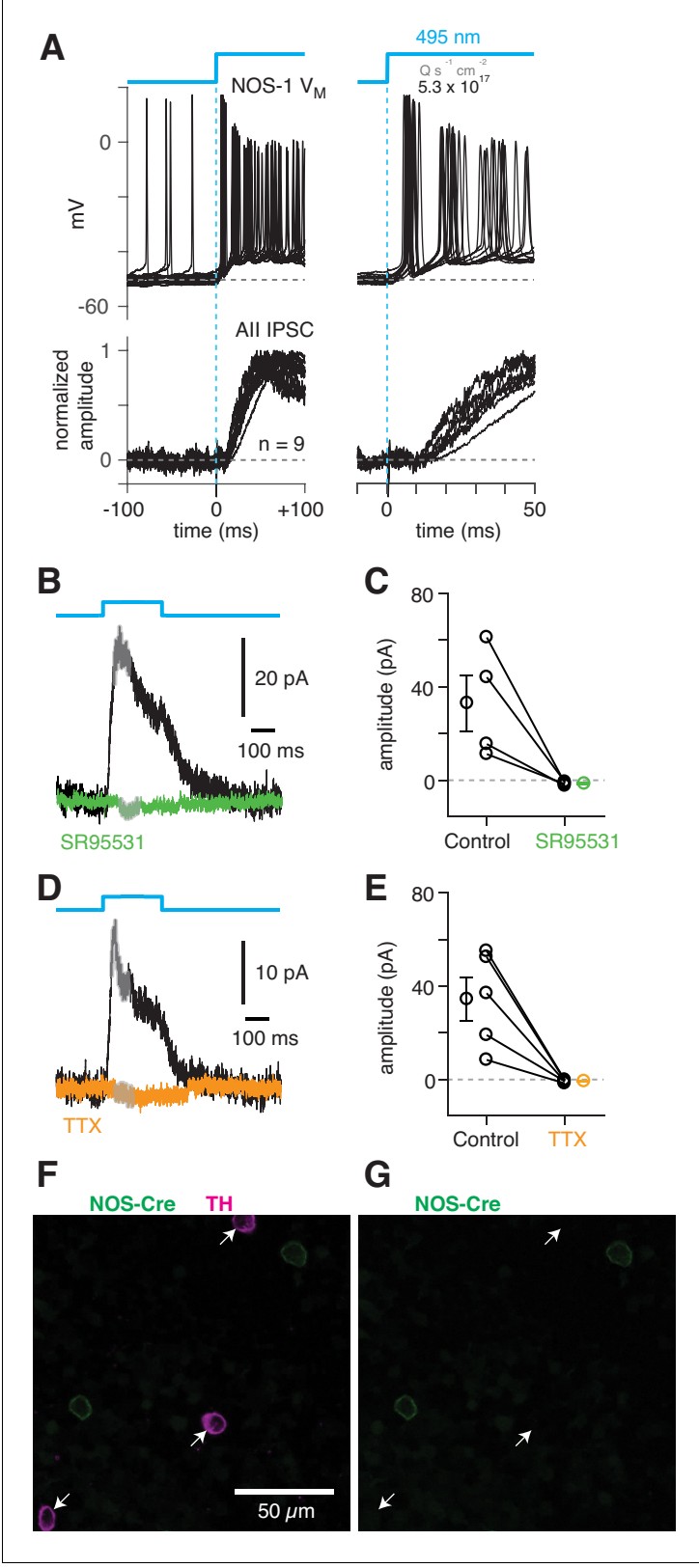

**Figure 8.** NOS-1 cells make synapses with AII amacrine cells. (**A**) Top/Left, optogenetic stimulation of a ChR2-expressing NOS-1 cell (of n = 3 total) in the nNOS-CreER::Ai32 retina responded with increased spike firing to blue light (n = 12 trials overlaid). Response was recorded in whole-mount retina in the presence of drugs to block

*Figure 8 continued on next page*

*Figure 8 continued*

photoreceptor-mediated inputs to retinal circuitry: DNQX (50 µM), D-AP5 (50 µM), L-AP4 (2 µM), and ACET (1 µM). Bottom/Left, the optogenetic stimulus evoked IPSCs ($V_{hold} = E_{cat}$) in AII ACs (n = 9 cells). Responses are normalized to the maximum amplitude, 39 ± 6 pA (measured over a 60–70 ms time window). Right, expanded version of traces at left. The initial spike in the NOS-1 cell occurred a few milliseconds after optogenetic stimulation (top), followed a few milliseconds later by the onset of IPSCs in the AII. (B) In AII ACs recorded under the conditions in (A), inhibitory current (measured over the gray region) was blocked by the GABA-A receptor antagonist SR95531 (50 µM). (C) Effect of SR95531 in a sample of AII ACs (n = 4 cells). Error bars indicate ± SEM across cells. (D) Same format as (B) with the sodium channel blocker TTX (1 µM). (E) Same format as (C) with TTX (n = 5 cells). (F) Confocal image of the inner nuclear layer of a retina from the nNOS-creER::Ai32 mouse. A tyrosine-hydroxylase (TH) antibody was used to label dopaminergic ACs (arrows), which did not overlap with Cre-expressing NOS+ ACs. (G) Same image as (F) without the TH labeling. None of the cells with TH immunolabeling (arrows) were eYFP+.

---

spiking AC (*Bloomfield and Xin, 2000*; *Figure 2D–G*). The NOS-1 AC satisfies all of the criteria for the mechanism generating the AII surround: it is a GABAergic (*Figure 8A,B*; *Zhu et al., 2014*), spiking ON AC (*Figure 5B,C*; *Figure 8C,D*; *Jacoby et al., 2018*) that provides the majority of its synaptic output to AIIs (*Figures 3* and *4*; *Table 2*). Furthermore, NOS+ ACs are necessary for generating TTX-dependent surround inhibition in AIIs (*Figure 7A,B*).

Our conclusion that the NOS-1 AC provides ~90% (*Table 1*) of the inhibitory input to the AII is based on the assumption—strongly supported by the anatomical evidence (*Figures 3* and *4*)—that a singlecell type is presynaptic to all of the inhibitory synapses on the distal dendrites of AIIs. Given that every inhibitory input to the distal dendrites of three AIIs analyzed arises from a process with a stereotyped pattern of synaptic output (*Figures 3C2* and *4A,B*), we assume they represent a single type. The alternative seems very unlikely: that there are multiple independent populations of amacrine cell, each of which has axons identical in appearance, running in exactly the same stratum of the IPL, providing the identical pattern of 80% output to AIIs and 20% output to RBs, and matching precisely the pattern of output of the two reconstructed cells (both of which share identical patterns of inputs as well as outputs). Indeed, it is established that individual retinal neuron types are defined by their highly stereotyped patterns of connectivity (*Briggman et al., 2011*; *Cohen and Sterling, 1990*; *Graydon et al., 2018*; *Hoggarth et al., 2015*). Further, the combination of wide-field axons in the ON laminae of the IPL and narrow-field dendrites in the OFF laminae of the IPL apparently is restricted to a single population of wide-field AC (*Lin and Masland, 2006*; *Zhu et al., 2014*). Finally, transcriptomic analysis supports the existence of two strongly expressing NOS+ ACs: NOS-1 and NOS-2 (*Yan et al., 2020*).

From our SBEM analysis, the multi-stratified AC presynaptic to the AII is predicted to be an ON cell that should be activated by dim scotopic stimuli owing to its input from RBs as well as from type 6 CBs, which are well-coupled to the AII network (*Grimes et al., 2014b*): our observations of the light responses of NOS-1 ACs (*Figure 6*), their synaptic connectivity (*Figures 8–9*), and the properties of surround suppression of GC responses to dim light stimuli (*Figure 2D*) collectively support the conclusion that the NOS-1 AC is the primary inhibitory neuron influencing the output of the AII network to rod-driven input. As well, it is notable that the NOS-

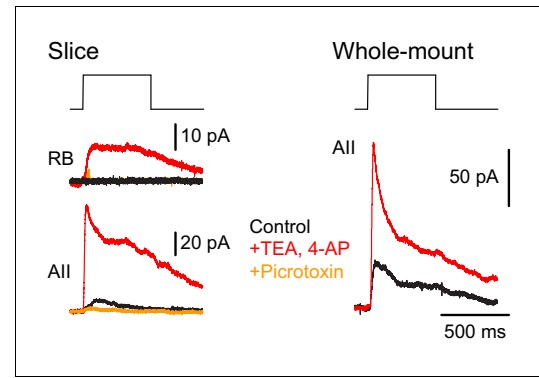

**Figure 9.** NOS-1 ACs make synapses with RBs. At left, recordings from a RB (top) and an AII (bottom) in a retinal slice demonstrate that optogenetic stimulation of cre-expressing cells in the nNOS-CreER::Ai32 retina evoked inhibitory currents ($V_{hold} = E_{cat}$). Potentiation of presynaptic depolarization with K channel blockers was necessary to elicit responses in RBs owing to the small number of presynaptic axons preserved in the 200 µm thin slice. At right, K channel blockers potentiate larger inhibitory currents ($V_{hold} = E_{cat}$) recorded in an AII and evoked by optogenetic stimulation of cre-expressing cells in a whole-mount preparation of nNOS-CreER::Ai32 retina.

1 AC appears to be one significant mechanism by which cone pathways can inhibit rod pathways (*Lauritzen et al., 2019*). And, given that the NOS-1 AC produces NO, which is thought to regulate electrical transmission between AIIs and ON cone bipolar cells (*Mills and Massey, 1995*), it is possible that synaptic inhibition of the AII is coupled with modulation of its electrical synapses. As noted above, we did not measure directly the NOS-1 AC response threshold to dim light due to the laser exposure required for cell targeting, but the response threshold of the NOS-1 AC is presumably among the most sensitive in the retina due to its input from RBs and type 6 CBs.

## Functional relevance

As luminance increases from visual threshold to 10–20 R*/rod/s, the gain of transmission at the RB→AII synapse is reduced (*Dunn et al., 2006*; *Dunn and Rieke, 2008*), and AIIs hyperpolarize; this hyperpolarization spreads to the terminals of type 6 CBs, which increases rectification while relieving activity-dependent synaptic depression at type 6→ON α GC synapses (*Grimes et al., 2014a*). The hyperpolarization of the AII in background light has been attributed to synaptic depression at the RB→AII synapse (*Grimes et al., 2014a*), but this hyperpolarization likely also depends on direct synaptic inhibition from the NOS-1 AC. The combined influence of these synaptic mechanisms would allow the RB pathway to remain functional at backgrounds as high as 250 R*/rod/s (*Ke et al., 2014*).

Thus, inhibitory input from the NOS-1 AC might represent a significant modulatory mechanism within the RB pathway and could be responsible for maintaining high-fidelity signaling through a range of background luminance at which differentiating signal from noise is a particular concern. More generally, because the type 6 CB provides input to the NOS-1 AC and is coupled electrically to the AII network that is inhibited by the NOS-1 AC, we propose that a type 6 CB→NOS-1 AC→AII AC→type 6 CB feedback circuit could maintain the rectifying nature of transmission at type 6 CB synapses by preventing excessive presynaptic depolarization across a range of lighting conditions. Testing this prediction would require the inactivation or ablation of NOS-1 cells while recording at various stages of the RB pathway. A limitation of the nNOS-CreER retina is that Cre is expressed in several cell types, including the NOS-2 cell, which is the main source of NO release within the retina (*Jacoby et al., 2018*). Improvements in the selective targeting of NOS-1 cells will be an important step in evaluating its function in retinal circuitry and behavior.

## Conclusion

We uncovered the dominant inhibitory input to the AII, the central neuron in the well-studied RB pathway of the mammalian retina. The NOS-1 AC is the spiking neuron responsible for generating the TTX-sensitive, GABAergic surround that modulates AII network function across a range of lighting conditions (*Bloomfield and Xin, 2000*; *Xin and Bloomfield, 1999*). Further, our anatomical analysis also explained the ON response of a multi-stratified AC with neurites and excitatory synaptic inputs in both the ON and OFF strata of the IPL. Our study thereby extends the classification of retinal neurons that receive ON input from *en-passant* axonal synapses made by ON bipolar cells in the OFF strata of the IPL, which also includes the dopaminergic AC and the intrinsically photosensitive ganglion cells (*Dumitrescu et al., 2009*; *Hoshi et al., 2009*; *Sabbah et al., 2017*). Our study demonstrates the utility of a targeted connectomic analysis coupled with neurophysiological investigation to neural circuit discovery.

# Materials and methods

**Key resources table**

| Reagent type (species) or resource | Designation | Source or reference | Identifiers | Additional information |
|---|---|---|---|---|
| Genetic reagent (*Mus musculus*) | C57BL/6J | Jackson Laboratory Stock #000664 | IMSR Cat# JAX000664, RRID:IMSR_JAX:000664 | Wild type mouse |
| Genetic reagent (*M. musculus*) | nNOS-CreER | Jackson Laboratory Stock #014541 | IMSR Cat# JAX014541, RRID:IMSR_JAX:014541 | Transgenic mouse: cre-driver line |

*Continued on next page*

*Continued*

| Reagent type (species) or resource | Designation | Source or reference | Identifiers | Additional information |
|---|---|---|---|---|
| Genetic reagent (*M. musculus*) | Ai14 | Jackson Laboratory Stock #007914 | IMSR Cat# JAX007914 RRID:IMSR_JAX:007914 | Transgenic mouse: cre-dependent tdTomato expression |
| Genetic reagent (*M. musculus*) | Ai32 | Jackson Laboratory Stock #024109 | IMSR Cat# JAX024109 RRID:IMSR_JAX:024109 | Transgenic mouse: cre-dependent ChR2/eYFP expression |
| Recombinant DNA reagent | AAV2/7m8-CAG-FLEX-DTA-WPRE-SC40pA | This paper | | Adeno-associated virus; $10^{13}$ GC/ml |
| Recombinant DNA reagent | pAAV-Syn-FLEX-rc[CoChR-GFP] | PMID:24509633 | Addgene #62724; RRID:Addgene_62724 | Plasmid |
| Recombinant DNA reagent | pAAV-CAG-fDIO-oG-WPRE-SV40pA | PMID:27149846 | Addgene #74291; RRID:Addgene_74291 | Plasmid |
| Recombinant DNA reagent | PGKdtabpA | PMID:9226440 | Addgene #13440; RRID:Addgene_13440 | Plasmid |
| Recombinant DNA reagent | AAV2/7m8 capsid | Gift of Dr. John Flannery, UC Berkeley; PMID:23761039 | Addgene #64839; RRID:Addgene_64839 | Plasmid |
| Antibody | anti-ChAT (Goat polyclonal) | EMD Millipore | Cat# AB144P; RRID:AB_2079751 | 1:200 dilution |
| Antibody | anti-LuciferYellow (Rabbit polyclonal) | ThermoFisher Scientific | Cat# A-5750; RRID:AB_2536190 | 1:2000 dilution |
| Antibody | anti-nNOS (Rabbit polyclonal) | ThermoFisher Scientific | Cat# 61–7000; RRID:AB_2313734 | 1:500 dilution |
| Antibody | anti-nNOS (Guinea pig polyclonal) | Frontier Institute | Cat# Af740; RRID:AB_2571816 | 1:2000 dilution |
| Antibody | anti-TH (Rabbit polyclonal) | EMD Millipore | Cat# AB152; RRID:AB_390204 | 1:1000 dilution |
| Antibody | anti-human/rat CRF serum (Rabbit polyclonal) | gift of Dr. Paul Sawchenko, Salk Institute | Code# PBL rC68 | 1:40,000 dilution |
| Antibody | anti-LHX9 (Guinea pig polyclonal) | gift of Dr. Jane Dodd, Columbia University | | 1:20,000 dilution |
| Antibody | Donkey Anti-Goat AlexaFluor 633 secondary | ThermoFisher Scientific | Cat # A-21082; RRID:AB_2535739 | 1:500 dilution |
| Antibody | Donkey Anti-Rabbit AlexaFluor 488 secondary | Jackson ImmunoResearch | Cat# 711-545-152; RRID:AB_2313584 | 1:500 dilution |
| Antibody | Donkey Anti-Rabbit Cy3 secondary | Jackson ImmunoResearch | Cat# 711-165-152; RRID:AB_2307443 | 1:500 dilution |
| Antibody | Donkey Anti-Rabbit Cy5 secondary | Jackson ImmunoResearch | Cat# 711-175-152; RRID:AB_2340607 | 1:500 dilution |
| Antibody | Donkey Anti-Guinea pig secondary | Jackson ImmunoResearch | Cat# 706-175-148; RRID:AB_2340462 | 1:500 dilution |
| Software, algorithm | ScanImage | http://scanimage.vidriotechnologies.com/ PMID:12801419 | RRID:SCR_014307 | Software for 2PLSM |

*Continued on next page*

*Continued*

| Reagent type (species) or resource | Designation | Source or reference | Identifiers | Additional information |
|---|---|---|---|---|
| Software, algorithm | ParaView | https://www.paraview.org/ | RRID:SCR_002516 | Software for visualization |
| Software, algorithm | IgorPro | https://www.wavemetrics.com/ | RRID:SCR_000325 | Software for data acquisition and analysis |
| Software, algorithm | PClamp | https://www.moleculardevices.com/ | RRID:SCR_011323 | Software for data acquisition and analysis |
| Software, algorithm | Knossos | https://knossos.app/ PMID:21743472 | RRID:SCR_003582 | Software for volumetric imaging analysis |
| Software, algorithm | ImageJ | https://ImageJ.net | RRID:SCR_003070 | Software for cell counting and image projection |

## TEM

An excised retina was fixed for one hour at room temperature with 2% glutaraldehyde in 0.15 M cacodylate buffer, washed in three changes of the same buffer, and postfixed with 1% osmium tetroxide in 0.15 M cacodylate containing 1.5% potassium ferrocyanide. A wash in three changes of distilled water followed the reduced osmium fixation and preceded an *en bloc* fix in 2% aqueous uranyl acetate. Dehydration in a graded series of ethanol (35% to 100%), and infiltration in a propylene oxide:epoxy resin series was followed by embedding and polymerization in epoxy resin. Thin sections were cut on a Reichert Ultracut E ultramicrotome, stained with 2% uranyl acetate and 0.2% lead citrate before being viewed and photographed on a Zeiss EM10 CA transmission electron microscope.

## SBEM analysis

Dataset k0725, a 50 × 210 × 260 µm block of fixed mouse retina imaged with voxel size 13.2 × 13.2 × 26 nm (*Ding et al., 2016*) was analyzed. Manual skeletonization and annotation were performed using Knossos [http://www.knossostool.org/; (*Helmstaedter et al., 2011*)]. Tracing began at RB terminals, which were easily identified based on their size and position within the inner plexiform layer (IPL) (*Mehta et al., 2014*; *Pallotto et al., 2015*). From RB dyad synapses, AIIs were traced, and then, from sites of synaptic input, ACs were traced. All skeletons and annotations were checked by two expert observers. Voxel coordinates were tilt-corrected and normalized to the positions of the ON and OFF SACs [per (*Helmstaedter et al., 2011*) identified in (*Ding et al., 2016*)]. Connectivity analysis was performed using custom-written Python scripts. Skeletons were visualized in Paraview (www.paraview.org).

## Electrophysiology

All animal procedures were approved by the Institutional Animal Care and Use Committees at Yale University or the University of Maryland. Experiments used offspring of nNOS-CreER and Ai32 mice. In nNOS-CreER mice (B6; 129S-Nos1$^{tm1.1(cre/ERT2)/Zjh}$/J; Jackson Laboratory #014541, RRID:IMSR_JAX:014541), expression of Cre recombinase is driven by endogenous *Nos1* regulatory elements (*Taniguchi et al., 2011*), and Cre expression was induced by tamoxifen (2 mg delivered on two consecutive days) administered by either IP injection or gavage at P31 (SD = 5.6 days), and at least 2 weeks before the experiment. Ai32 mice (B6;129S-Gt(ROSA)26Sor$^{tm32(CAG-COP4*H134R/EYFP)Hze}$/J; Jackson Laboratory #024109, RRID:IMSR_JAX:024109) express a Cre-dependent channelrhodopsin-2 (ChR2)/enhanced yellow fluorescent protein (eYFP) fusion protein (*Madisen et al., 2012*). Ai14 mice (B6;129S6-Gt(ROSA)26Sor$^{tm14(CAG-tdTomato)Hze}$/J; Jackson Laboratory #007908), used for some experiments with DTA virus, express a Cre-dependent tdTomato protein (*Madisen et al., 2010*). Mice studied were heterozygous for the Cre allele and either the Ai32 or the Ai14 reporter allele.

A mouse aged between ~2 and 4 months was dark adapted for 1 hr, and following death, the eye was enucleated and prepared for recording in Ames medium (Sigma) under infrared light using night-vision goggles connected to a dissection microscope (*Park et al., 2015*). In the recording chamber, the retina was perfused (~4–6 ml/min) with warmed (31–34˚C), carbogenated (95% $O_2$-5% $CO_2$) Ames' medium (light response and optogenetic experiments in whole-mount retina). The retina was imaged using a custom-built two-photon fluorescence microscope controlled with ScanImage software (RRID:SCR_014307; *Borghuis et al., 2013*; *Borghuis et al., 2011*; *Pologruto et al., 2004*). Fluorescent cells were targeted for whole-cell patch clamp recording with a Coherent Ultra II laser tuned to 910 nm (*Park et al., 2015*). For optogenetic experiments in retinal slices, dissection was performed in normal room light, and the retina was maintained in artificial CSF as described previously (*Jarsky et al., 2010*).

Electrophysiological measurements were made by whole-cell recordings with patch pipettes (tip resistance 4–11 MΩ). Membrane current or potential was amplified, digitized at 10–20 kHz, and stored (MultiClamp 700B amplifier; ITC-18 or Digidata 1440A A-D board) using either pClamp 10.0 (Molecular Devices) or IGOR Pro software (Wavemetrics). For light-evoked responses and optogenetic experiments in whole-mount retina, pipettes contained (in mM): 120 Cs-methanesulfonate, 5 TEA-Cl, 10 HEPES, 10 BAPTA, 3 NaCl, 2 QX-314-Cl, 4 ATP-Mg, 0.4 GTP-$Na_2$, and 10 phosphocreatine-$Tris_2$ (pH 7.3, 280 mOsm). For optogenetic experiments in retinal slices, pipettes contained (in mM): 90 Cs-methanesulfonate, 20 TEA-Cl, 1 4-AP, 10 HEPES, 1 BAPTA, 4 ATP-Mg, 0.4 GTP- $Na_2$, and 8 phosphocreatine-$Tris_2$.

Either Lucifer Yellow (0.05–0.1%) or red fluorophores (sulfarhodamine, 10 µM or Alexa 568, 60 µM) were added to the pipette solution for visualizing the cell. All drugs used for electrophysiology experiments were purchased from Tocris Biosciences, Alomone Laboratories or Sigma-Aldrich. Excitatory and inhibitory currents were recorded at holding potentials near the estimated reversal for either $Cl^-$ ($E_{Cl}$, −67 mV) or cations ($E_{cation}$, +5 mV), after correcting for the liquid junction potential (−9 mV). Series resistance (~10–80 MΩ) was compensated by up to 50%. Following the recording, an image of the filled cell was acquired using 2PLSM.

Light stimuli were presented using a modified video projector [peak output, 397 nm; full-width-at-half-maximum, 20 nm (*Borghuis et al., 2014*; *Borghuis et al., 2013*) focused onto the retina through the microscope condenser. Stimuli were presented within a 4 × 3 mm area on the retina. Stimuli included contrast-reversing spots of variable diameter to measure spatial tuning (*Zhang et al., 2012*). For some experiments, stimuli were presented with 1 Hz temporal square-wave modulations (100% Michelson contrast) relative to a background of mean luminance that evoked ~$10^4$ photoisomerizations (R*) $cone^{-1}$ $s^{-1}$ (*Borghuis et al., 2014*). In other experiments, stimuli were spots of dim light (4–40 R* $rod^{-1}$ $s^{-1}$) of varying diameter presented on a dark background. We did not attempt to use dimmer stimuli owing to concerns about maintaining the retina in a completely dark-adapted state over the duration of the experiment: for example targeting AII ACs for recording in the whole-mount retina required exposure to infrared light sufficient to stimulate rods, albeit weakly, over a prolonged period (>10 min) and therefore precluded complete dark adaptation of the retina.

## Optogenetics

ChR2-mediated responses were recorded in the presence of drugs to block conventional photoreceptor-mediated light responses. Recordings were made in a cocktail of (in µM): L-AP4 (20); either UBP310 (50) or ACET (1-5); DNQX (50-100); and D-AP5 (50–100) (*Park et al., 2015*). ChR2 was activated by a high-power blue LED ($\lambda_{peak}$, 450 or 470 nm; maximum intensity of ~$5\times10^{17}$ photons $s^{-1}$ $cm^{-2}$) focused through the condenser onto a square (220 µm per side) area as described previously (*Park et al., 2015*).

## Construction and production of recombinant AAV

To generate an AAV vector backbone, we modified two plasmids procured from Addgene (#62724 and #74291). We digested plasmid #74291 with BamHI, treated with Klenow fragment, digested again with HindIII, and kept the vector backbone. The insert part of plasmid #62724 was excised by digesting with EcoRI, treated with Klenow fragment, and digested again with HindIII. The excised fragment was ligated into the vector backbone from plasmid #74291. Then, we amplified the DTA

sequence by PCR, created KpnI and NheI restriction sites at each end, and subcloned the PCR products into a newly generated AAV vector. The final construct contained DTA sequence in reverse orientation surrounded by two nested pairs of incompatible loxP sites (pAAV-CAG-FLEX-NheI-DTA-KpnI-WPRE-SV40pA). The plasmid carrying DTA was obtained from Addgene (#13440).

Virus production was based on a triple-transfection, helper-free method, and the virus was purified as described previously (*Byun et al., 2019*; *Park et al., 2015*), except that we used a plasmid carrying AAV2/7m8 capsid (gift from Dr. John Flannery, University of California at Berkeley). The titer of the purified AAVs was determined by quantitative PCR using primers that recognize WPRE; the concentrated titers were >$10^{13}$ viral genome particles/ml in all preparations. Viral stocks were stored at −80°C.

## Histology

For immunohistochemistry, animals were perfused at age 1–12 weeks. The retinas were dissected and fixed with 4% paraformaldehyde for 1 hr at 4°C. For whole-mount staining, retinas were incubated with 6% donkey serum and 0.5% triton X-100 in PBS for 1 hr at room temperature; and then incubated with 2% donkey serum and 0.5% triton X-100 in PBS with primary antibodies for 1–4 days at 4°C, and with secondary antibodies for 1–2 hr at room temperature. For morphological analysis of recorded cells, the retina was fixed for 1 hr at room temperature and reacted as described previously (*Manookin et al., 2008*).

Primary antibodies were used at the following concentrations: goat anti-ChAT (1:200, Millipore AB144P, RRID:AB_2079751), rabbit anti-Lucifer Yellow (1:2000, ThermoFisher Scientific A-5750, RRID:AB_2536190), rabbit anti-nNOS (1:500, ThermoFisher Scientific 61–7000, RRID:AB_2533937), guinea pig anti-nNOS (1:2000, Frontier Institute Af740, RRID:AB_2571816), rabbit anti-TH (1:1000, Millipore AB152, RRID:AB_390204), rabbit anti-human/rat CRF serum (1:40,000, code #PBL rC68; gift of Dr. Paul Sawchenko, Salk Institute), guinea pig anti-LHX9 (1:20,000, gift of Dr. Jane Dodd, Columbia University). Rabbit anti-nNOS was used in all cases of nNOS immunolabeling except for *Figure 6F and G*, in which case guinea pig anti-nNOS was used. Secondary antibodies (applied for 2 hr) were conjugated to Alexa Fluor 488, Cy3 and Cy5 (Jackson ImmunoResearch or ThermoFisher Scientific) and diluted at 1:500.

## Confocal imaging

Confocal imaging was performed using Zeiss laser scanning confocal microscopes (510, 710, or 800 models). For filled cells, a whole-mount image of the dendritic tree was acquired using a 20X air objective (NA = 0.8); in some cases, multiple images were combined as a montage. A high-resolution z-stack of the ChAT bands (i.e. cholinergic starburst AC processes, labeled by the ChAT antibody) and the filled amacrine or ganglion cell was obtained to determine their relative depth in the IPL using a 40X oil objective (NA = 1.4). Analysis of nNOS, Lhx9, CRH and TH antibody labeling was performed either with the 40X oil objective or the 20X air objective. Cells were counted using ImageJ (RRID:SCR_003070).

## Acknowledgements

Supported by EY017836 to JHS, EY014454 and EY029323 to JBD, EY029820 to I-JK, P30-EY026878 to Yale University, a grant from the Knights Templar Eye Foundation to SJHP and an unrestricted grant from Research to Prevent Blindness to Yale University. We thank Tim Maugel and the Laboratory for Biological Ultrastructure (Department of Biology, UMD) for assistance with transmission electron microscopy, Alexander Baden for assistance with 3D visualization, and Kacie Furcolo and Adit Sabnis for assistance with SBEM dataset skeletonization. We thank Cole Graydon for comments on the manuscript.

## Additional information

### Funding

| Funder | Grant reference number | Author |
|---|---|---|
| National Eye Institute | EY017836 | Joshua H Singer |
| National Eye Institute | EY014454 | Jonathan B Demb |
| National Eye Institute | EY029820 | In-Jung Kim |
| National Eye Institute | P30-EY026878 | In-Jung Kim<br>Jonathan B Demb |
| Research to Prevent Blindness | | In-Jung Kim<br>Jonathan B Demb |
| National Eye Institute | EY029323 | Jonathan B Demb |
| Knights Templar Eye Foundation | | Silvia JH Park |

The funders had no role in study design, data collection and interpretation, or the decision to submit the work for publication.

### Author contributions

Silvia JH Park, Conceptualization, Formal analysis, Supervision, Funding acquisition, Investigation, Writing - original draft, Project administration, Writing - review and editing; Evan E Lieberman, Jiang-Bin Ke, Formal analysis, Investigation; Nao Rho, Padideh Ghorbani, Software, Investigation; Pouyan Rahmani, Investigation; Na Young Jun, Kevin L Briggman, Resources, Software, Investigation, Methodology; Hae-Lim Lee, Resources, Supervision, Funding acquisition; In-Jung Kim, Conceptualization, Supervision, Funding acquisition, Writing - original draft, Writing - review and editing; Jonathan B Demb, Conceptualization, Resources, Supervision, Funding acquisition, Writing - original draft, Writing - review and editing; Joshua H Singer, Conceptualization, Resources, Formal analysis, Supervision, Funding acquisition, Investigation, Writing - original draft, Project administration, Writing - review and editing

### Author ORCIDs

Na Young Jun http://orcid.org/0000-0002-8841-3947
Kevin L Briggman http://orcid.org/0000-0003-2946-4661
Jonathan B Demb https://orcid.org/0000-0003-2227-9041
Joshua H Singer https://orcid.org/0000-0002-0561-2247

### Ethics

Animal experimentation: All animal procedures were approved by the Institutional Animal Care and Use Committees at Yale University or the University of Maryland.

### Decision letter and Author response

Decision letter https://doi.org/10.7554/eLife.56077.sa1
Author response https://doi.org/10.7554/eLife.56077.sa2

## Additional files

### Supplementary files

• Source code 1. A compressed file containing Python scripts used to parse Knossos XML files containing skeletons and annotations: see 'tutorial.py' for descriptions.

• Source data 1. Skeletons of AIIs and other ACs.

- Source data 2. Skeletons of ON CBs, NOS-1 ACs, and one ON SAC.

- Source data 3. Skeletons of ON and OFF SACs.

- Transparent reporting form

## Data availability

SBEM dataset k0725 is was published in Ding et al. 2016. The dataset is available in the repository at https://webknossos.org. The dataset also can be provided on a hard drive by arrangement with Dr. Kevin L Briggman.

The following previously published dataset was used:

| Author(s) | Year | Dataset title | Dataset URL | Database and Identifier |
|---|---|---|---|---|
| Ding H, Smith RG, Poleg-Polsky A, Diamond JS, Briggman KL | 2016 | k0725 | https://webknossos.org/datasets/Demo_Organization/110629_k0725/view | webKnossos, 110629_k0725 |

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
