## [Decision Letter]

Thank you for submitting your article "Connectomic analysis reveals an interneuron with an integral role in the retinal circuit for night vision" for consideration by *eLife*. Your article has been reviewed by three peer reviewers, including Fred Rieke as the Reviewing Editor and Reviewer #1, and the evaluation has been overseen by Ronald Calabrese as the Senior Editor. The following individuals involved in review of your submission have agreed to reveal their identity: Greg D Field (Reviewer #2); Daniel Kerschensteiner (Reviewer #3).

As you will see from the individual reviews, all three of the reviewers were quite enthusiastic about the work and appreciated the breadth of approaches used to investigate this amacrine type and its role. We also had a discussion after submitting our reviews about the request for additional experiments, which is common across the reviews as you will see. We are all in agreement that, given the uncertainties of doing those experiments, you should view those as optional rather than mandatory – and specifically you should not wait to resubmit if doing extra experiments is impossible at this time.

Reviewer #1:

This paper uses an impressive array of techniques to identify an amacrine cell type involved in spatial processing in the retinal rod bipolar pathway. The cell is initially identified using electron microscopy to find conventional (non-ribbon) synapses onto AII amacrine cells and the processes of the presynaptic cells that make those synapses. Two complete cells are then reconstructed and based on their morphology a Cre line identified with a morphologically-similar set of cell labeled. The labeled cells (NOS-1 amacrine cells) have properties consistent with those expected from the em reconstructions. Nice direct experiments (optogenetics and cell elimination) confirm that the NOS-1 cells indeed provide inhibitory input to AII amacrines. The findings presented provided good support for the conclusions reached. I have several questions and suggestions about places the paper could be strengthened:

Subsection “Functional relevance”: Was it not possible to probe the NOS-1 cells at lower light levels? It would be quite interesting to see the contributions of rod and cone input to their responses given the proposed circuitry and functional role.

The assumption that all conventional synaptic input to the NOS-1 cell is inhibitory should be qualified given the presence of amacrine cells that release glutamate, and non-ribbon synapses made by bipolar cells.

Quantification. Several anatomical observations would be stronger with some quantification. This includes the proximity of the AC-AII synapses and the dendritic branching angle (subsection “Anatomical identification of inhibitory synaptic inputs to AIIs”).

Subsection “The NOS-1 AC is the predominant NOS-expressing AC in the ganglion cell layer and is distinct from CRH-expressing ACs”: how does the sustained ON response uniquely identify NOS-1 cells? Specifically, could there be another cell type with a sustained ON response that is mixed in? To establish this definitively it seems you would need to fill the recorded cells and see if they share similar morphology. Related, do the brightly labeled cells have NOS-1 morphologies?

Subsection “Propagation of the AII surround to downstream ganglion cells”: surrounds inherited from the AII might be expected to be similar across RGC types. This does not appear to be the case, as the surround in the OFF α looks somewhat stronger than the other two cells. It would be helpful to fit the dependence of the response on stimulus diameter to compare parameters of a difference of gaussian model and see if they are consistent.

Reviewer #2:

As the author's point out, the rod bipolar –> AII amacrine cell circuit is one of the most studied circuits in the mammalian retina. It is also an immensely important circuit as it serves to reliably and efficiently process visual input under conditions when photons are scarce. Given all the work that has been performed previously on this circuit, the bar is rather high to make a substantive and transformative contribution to our understanding of it. This manuscript, in my view, sails over that bar. The study is an impressive combination of high-throughput anatomical reconstructions, clever utilization of transgenic mouse lines, and careful physiological measurements to identify a new and important element of the retina's primary circuit for night vision. In particular, the manuscript identifies the NOS-1 amacrine cell as the primary cell type that provides a receptive field surround (via GABAergic lateral inhibition) to AII amacrine cells. From a scientific perspective, I could not find significant issues with the logic or execution of the experiments described in this manuscript. It is one of the best papers I have reviewed in some time.

I do think the authors could make more of an effort to connect to a wider audience in the Abstract and Introduction. The paper is a straightforward read for a retinal neuroscientist, but I'm not sure a neuroscientist outside of that field will understand what is really interesting and important here. For example, the first two sentences of the Abstract build almost zero interest and give little insight to the core problem or why understanding this circuit is interesting. Pitching the paper as “understanding where the surround under scotopic conditions comes from” would at least pull in a broader community of vision scientists who would understand why that problem is important. Similarly, in the Introduction, the authors don't really cue the reader into the question they are tackling until the end of the 4th paragraph. I don't think a full exposition of the rod circuit is necessary before letting the reader know what question the paper is answering, and the current approach seems likely to put off casual readers. That said, leaving the manuscript “as is” would already put it in the top ~5% of manuscripts that I've read.

Reviewer #3:

In this study, Park et al. identify nNOS1 amacrine cells as a component of the retinal circuit that processes low-light information. The authors first show that the surround inhibition of AII amacrine cells, which relay low-light signals from rod bipolar to cone bipolar and ganglion cells, relies on spiking amacrine cells. They then trace the inhibitory inputs to AII amacrine cells in a previously published serial blockface scanning electron microscopy (SBEM) volume and find that many inputs are provided by axonal processes that send output to AIIs but receive little or no input in the volume. However, 2 of the 63 traced processes are connected to cell bodies within the volume. Tracing of these cells reveals bistratified arbors (large ON, small OFF) in the inner plexiform layer. These arbors receive input from ON cone bipolar cells (mostly type 6) and rod bipolar cells and from other amacrine cells near the cell body and provide output in their periphery. Given morphological similarities, the authors hypothesize that the respective cells are nNOS1 amacrine cells. They target nNOS1 amacrine cells in nNOS-CreER mice and show (1) that they are spiking amacrine cells and (2) that in spite of their bistratified arbors excitatory inputs and spikes are elicited solely by ON stimuli, consistent with SBEM reconstructions. The authors confirm functional connectivity between nNOS1 and AII amacrine cells by optogenetics. Finally, they use viral expression of diphtheria toxin to ablate nNOS1 amacrine cells and show that this abolishes surround inhibition of AII amacrine cells.

Overall, this is an excellent study, impressive in its scope and technical ability. It combines electrophysiological recordings with SBEM reconstructions, optogenetic mapping, and a cell-type-specific perturbation to identify a new component of a well-studied retinal circuit with known behavioral functions. Experiments are performed to a high technical standard and results are presented clearly. However, the presentation of the anatomical data could be enriched, larger stimuli should be included when probing the surround inhibition of AII amacrine cells, and I encourage the authors to take advantage of their cell-type-specific perturbation to test the impact of nNOS1 amacrine cells on dim light signals of ganglion cells (i.e., the output from the retina to the brain).

Specific comments:

1) Surround inhibition of AII amacrine cells in control conditions (Figure 2) is tested only with small stimuli. This is a mismatch to the stimuli used for ganglion cells, nNOS1 amacrine cells, and AII cells after the deletion of nNOS1 amacrine cells. The authors should include larger stimuli in their experiments in Figures 2A-D, to allow for better comparisons between datasets.

2) The authors show that the removal of nNOS1 amacrine cells by viral expression of diphtheria toxin reduces the surround inhibition of AII amacrine cells. To elucidate the significance of nNOS1 amacrine cells to the output signals of the retina, I encourage the authors to test the effects of nNOS1 removal on spike response of ON α, OFF α, and OFF δ cells. This would boost the significance of the study.

3) I suggest changing the order in which optogenetic and diphtheria toxin experiments are presented. It seems more natural to me to follow the anatomical discovery of connections between nNOS1 and AII amacrine cells with their functional confirmation (i.e., optogenetics) and then to move on to the perturbation that elucidates the functional significance of nNOS1 – AII amacrine cell connections.

4) For the synaptic constellations in Figures 4D and 4E, it would be great if the authors could include supplementary videos scrolling through successive planes.

5) I was slightly confused by the inclusion of ON SACs as a laminar marker in Figure 4C and suggest switching to the same gray ON and OFF SAC bands as in Figure 4A.

6) In a supplement for Figure 4, it would be great if the authors could show en face views of the arrays of the different bipolar cell types connected to each nNOS1 amacrine cells (including only bipolar cells with synapses onto the respective target).

---

## [Author Response]

Reviewer #1:[…] I have several questions and suggestions about places the paper could be strengthened:Subsection “Functional relevance”: Was it not possible to probe the NOS-1 cells at lower light levels? It would be quite interesting to see the contributions of rod and cone input to their responses given the proposed circuitry and functional role.

The main difficulty with this experiment is that we have to identify fluorescent reporter expressing NOS-1 cells by 2PLSM using an IR laser, the light from which alters the responses of rods and shifts their response thresholds, as a number of groups have shown. Therefore, we decided to work at a light level above absolute rod threshold but still within the range where rod bipolar cells are active. This is now explained in the Discussion (subsection “The NOS-1 AC is the dominant inhibitory input to the AII and generates the AII surround”) and the Materials and methods section.

The assumption that all conventional synaptic input to the NOS-1 cell is inhibitory should be qualified given the presence of amacrine cells that release glutamate, and non-ribbon synapses made by bipolar cells.

We added some discussion of glutamatergic amacrine cells, though, as we note, it is unlikely that the well-studied VGLUT3 cell provides synapses to the NOS-1 AC. As well, we make the point that we followed the neurites of putative AC inputs over some short distance to confirm their identities as ACs (subsection “Anatomical assessment of an AC circuit presynaptic to the AII”).

Quantification. Several anatomical observations would be stronger with some quantification. This includes the proximity of the AC-AII synapses and the dendritic branching angle (subsection “Anatomical identification of inhibitory synaptic inputs to AIIs”).

We calculated the distance between each AC–>AII synapse and the nearest RB–>AII synapse (on the same AII). The resulting distribution is illustrated in Figure 3C4. We note that 50% of the AC–>AII synapses are within 2.26 µm of a RB–>AII synapse, indicating that the inhibitory AC input is positioned to modulate the AII response to excitatory input from the RB (subsection “Anatomical identification of inhibitory synaptic inputs to AIIs”). Also, we deleted any mention of branching angle from the revised text because we do not have enough observations to make any quantitative assessment of the significance of the observation.

Subsection “The NOS-1 AC is the predominant NOS-expressing AC in the ganglion cell layer and is distinct from CRH-expressing ACs”: how does the sustained ON response uniquely identify NOS-1 cells? Specifically, could there be another cell type with a sustained ON response that is mixed in? To establish this definitively it seems you would need to fill the recorded cells and see if they share similar morphology. Related, do the brightly labeled cells have NOS-1 morphologies?

We didn’t observe any diversity in the morphology of these cells: all had small field of dendrites in the OFF layer and branching axons in the ON layer. This is mentioned in the revised text (subsection “The NOS-1 AC is a spiking ON cell”).

Subsection “Propagation of the AII surround to downstream ganglion cells”: surrounds inherited from the AII might be expected to be similar across RGC types. This does not appear to be the case, as the surround in the OFF α looks somewhat stronger than the other two cells. It would be helpful to fit the dependence of the response on stimulus diameter to compare parameters of a difference of gaussian model and see if they are consistent.

We are going to decline, respectfully, to take this suggestion. For this purpose, we think the difference-of-Gaussians model to be too poorly constrained (fitting diameter-response curves with four parameters: peak and space constant of center and surround) to be useful. Furthermore, the recorded IPSCs reflect all inhibitory input to the OFF α and δ cell types, not just the AII input, and therefore diversity across ganglion cells is expected: we would not expect identical tuning curves unless the synaptic inputs to the cells were identical, which they are not. We commented on this point (subsection “Propagation of the AII surround to downstream ganglion cells”).

Reviewer #2:[…]I do think the authors could make more of an effort to connect to a wider audience in the Abstract and Introduction. The paper is a straightforward read for a retinal neuroscientist, but I'm not sure a neuroscientist outside of that field will understand what is really interesting and important here. For example, the first two sentences of the Abstract build almost zero interest and give little insight to the core problem or why understanding this circuit is interesting. Pitching the paper as “understanding where the surround under scotopic conditions comes from” would at least pull in a broader community of vision scientists who would understand why that problem is important. Similarly, in the Introduction, the authors don't really cue the reader into the question they are tackling until the end of the 4th paragraph. I don't think a full exposition of the rod circuit is necessary before letting the reader know what question the paper is answering, and the current approach seems likely to put off casual readers. That said, leaving the manuscript “as is” would already put it in the top ~5% of manuscripts that I've read.

We thoroughly revised the Abstract and Introduction accordingly.

Reviewer #3:Specific comments:1) Surround inhibition of AII amacrine cells in control conditions (Figure 2) is tested only with small stimuli. This is a mismatch to the stimuli used for ganglion cells, nNOS1 amacrine cells, and AII cells after the deletion of nNOS1 amacrine cells. The authors should include larger stimuli in their experiments in Figures 2A-D, to allow for better comparisons between datasets.

While the paper was under review, we had the opportunity to collect additional data from a more suitable control group (Cre-negative animals that had been injected with AAV and tamoxifen; and from the same breeding cage as the Cre-positive/DTA animals). We also used the same range of spot sizes. The new data are presented in Figure 7. The responses at the largest three spot sizes was significantly smaller in the DTA recordings compared to the control recordings. We also updated all of the control cell counting data from these new retinas (n = 6; Figure 7).

2) The authors show that the removal of nNOS1 amacrine cells by viral expression of diphtheria toxin reduces the surround inhibition of AII amacrine cells. To elucidate the significance of nNOS1 amacrine cells to the output signals of the retina, I encourage the authors to test the effects of nNOS1 removal on spike response of ON α, OFF α, and OFF δ cells. This would boost the significance of the study.

We are not able to collect this data. Further, we think that this experiment would be complicated by the fact that the nNOS-CreER retina labels multiple cell types, and so the DTA will necessarily ablate NOS-2 cells, for example, which are the main source of NO release in the retina. The total effect of ablating all NOS+ cells could have effects beyond the elimination of the NOS-1 –> AII synapse. For this reason, we think a full evaluation of the role of NOS-1 cells in retinal function (and behavior) will require more selective genetic targeting of the NOS-1 cell. We now make this point in the Discussion section (subsection “Functional relevance”).

3) I suggest changing the order in which optogenetic and diphtheria toxin experiments are presented. It seems more natural to me to follow the anatomical discovery of connections between nNOS1 and AII amacrine cells with their functional confirmation (i.e., optogenetics) and then to move on to the perturbation that elucidates the functional significance of nNOS1 – AII amacrine cell connections.

We are going to decline, respectfully, to do this. We think that the optogenetic experiments are a better indicator of the function of the circuit than are the DTA experiments.

4) For the synaptic constellations in Figures 4D and 4E, it would be great if the authors could include supplementary videos scrolling through successive planes.

We have created two supplementary videos.

5) I was slightly confused by the inclusion of ON SACs as a laminar marker in Figure 4C and suggest switching to the same gray ON and OFF SAC bands as in Figure 4A.

We revised the figure accordingly.

6) In a supplement for Figure 4, it would be great if the authors could show en face views of the arrays of the different bipolar cell types connected to each nNOS1 amacrine cells (including only bipolar cells with synapses onto the respective target).

We revised the figure accordingly.